# Robust small area estimation for unit level model with density power divergence

**Xijuan Niu**[1,2]*, **Zhiqiang Pang**[1], **Zhaoxu Wang**[1,2]

**1** Department of Statistics, Lanzhou University of Finance and Economics, Lanzhou, Gansu, China,
**2** Department of Statistics Mathematics and Statistics, Qinghai Normal University, Xining, Qinghai, China

* 2014107@qhnu.edu.cn

## Abstract

Unit level model is one of the classical models in small area estimation, which plays an important role with unit information data. Empirical Bayesian(EB) estimation, as the optimal estimation under normal assumption, is the most commonly used parameter estimation method in unit level model. However, this kind of method is sensitive to outliers, and EB estimation will lead to considerable inflation of the mean square error(MSE) when there are outliers in the responses $y_{ij}$. In this study, we propose a robust estimation method for the unit-level model with outliers based on the minimum density power divergence. Firstly, by introducing the minimum density power divergence function, we give the estimation equation of the parameters of the unit level model, and obtain the asymptotic distribution of the robust parameters. Considering the existence of tuning parameters in the robust estimator, an optimal parameter selection algorithm is proposed. Secondly, empirical Bayesian predictors of unit and area mean in finite populations are given, and the MSE of the proposed robust estimators of small area means is given by bootstrap method. Finally, we verify the superior performance of our proposed method through simulation data and real data. Through comparison, our proposed method can can solve the outlier situation better.

## 1 Introduction

In sampling estimation, due to the challenge of small sample or even no sample, small area estimation(SAE) has received unanimous favor from statisticians [1–4]. Compared with the traditional direct estimation method, the small area estimation can solve the small sample problem better by "borrowing strength" from the auxiliary information. In practice, the small area estimation method is widely used in population statistics [1], medical statistics [2], agricultural statistics [5, 6], poverty rate estimation [3, 7] and other fields. In terms of theoretical research, the theory of small area estimation has also been fully developed, forming a relatively complete theoretical system, [8] describe the basic theory of small area; [9] introduces the theory of small area estimation of several kinds of mixed models. For comprehensive overviews of small area estimation, see [8–11].

Among the SAE methods, the model-based SAE method has received more attention from statisticians. Although direct estimation can give an unbiased estimator of the target variable, due to the small sample size, direct estimator using only the original data is not reliable. The

**Data Availability Statement:** This data can be obtained from the R package "rsae", https://github.com/tobiasschoch/rsae.

**Funding:** This work has been supported in part by the Excellent Graduate Research Project in Gansu

Province: "Analysis of Household Surveys Based on Small Area Estimation" (2021CXZX-698) and the the Young Scientists Fund of Qinghai Normal University "Application Research of Minimum Density Power Divergence in Robust Small Area Estimation" (KJQN2022014). The funders had supported the data collection and analysis, decision to publish, preparation of the manuscript of this study.

**Competing interests:** The authors have declared that no competing interests exist.

small sample size of raw data can be overcome by statistical models using auxiliary variables. As one of the basic models of small area estimation, unit level model can deal with the estimation of target variables at each unit level in a small area and calculate the corresponding area level values from unit data. Due to the limitations of data collection, acquisition of auxiliary variables, and model calculation, unit level model is not as concerned by scholars as area level model. If observation data and auxiliary information can be obtained at the unit level, the establishment of the unit level model is a better choice for SAE. [5] use the nested error regression(NER) model to estimate crop area at county level based on sampling data and satellite data. [12] generalizes the NER model and discusses the estimators under the generalized linear mixed model by using the hierarchical Bayes(HB) method. Empirical best linear unbiased estimator (EBLUP) is the most widely used method in model-based SAE, which can solve the estimation problem of mixed linear models well. When the observed variables are dichotomous variables or counting variables, the empirical Bayes (EB) method is more widely used, for example, [1, 2] have been mentioned. In the basic unit level model, the errors of individual and area-specific random effect are assumed to follow the normal distribution. However, [13] points out that this assumption is difficult to be verified in practice, which means that the traditional EB method is not reliable in the presence of outliers. Meanwhile, in practical application, due to the small sample size, outliers are are very common, which will cause large estimation deviation in the traditional estimation method. In this paper, we focus on unit level models whose random effects have different skewed distributions or observations in the model have outliers, and propose robust estimation methods for such models.

The existence and influence of outliers in sampling estimation have been studied for a long time. [14] mentioned that outliers are a fact of life for any survey. [15] discussed how outliers can affect shrinkage estimators, even a single outlier may lead all the small area estimates to collapse to their corresponding direct estimates. At present, there are several common methods to deal with outlier observations [14, 16–18]. In the first method, the outliers are deleted directly and the remaining observations are used for estimation and prediction. Obviously, this method may be feasible to delete a few outlier observations when the sample size is very large. However, when the sample size is small, the method will not only cause the loss of information, but also lead to the deviation of the estimation from the real value. In the second method, non-outlier observations are used instead of outliers for estimation, and robust projection is used for robust prediction in the unsampled population. However, [16] states that an observation cannot be considered unique if it is accurately captured, and there is no reason to think that outlier observations cannot be included in an unsampled population. [14] have used robust projection method to construct robust small area estimator, and obtained MSE estimation for robust predictors. [3] proposed using global-local shrinkage priors for modeling random effects that allow potential outliers in the areal effect. In the other method, the influence of outliers on estimation results can be reduced by constructing robust estimators insensitive to outlier observations, which is also the main method concerned by scholars.

The early research on robust estimation of SAE can be traced back to the mean robust estimators of strata means, which proposed by [19], with small area means being a special case. [15] used a hierarchical Bayes (HB) framework to study the effect of outliers in $v_i$. The HB estimators based on long-tailed distributions (such as, $t$ and Cauchy) are more robust to outlying area-specific error $v_i$ than the estimators based on normal distribution. [20] research results indicate that the use of a $t$–distribution with small $k$ can diminish the effect of outliers in the sense that more weight is given to the direct estimator $\widehat{\boldsymbol{\theta}}_i$ than in the case of normal area-specific error. However, these two methods only consider the robust estimation in special cases, and do not discuss the properties of the estimation parameter. [17] introduced robust SAE

based on quantiles, which used M-quantiles to offer an alternative to the modeling of between area variation. [21] proposed a robust Bayes predictor for the FH model that can overcome the over-shrink caused by outlier observations, but the influence of outliers on the model coefficients is not taken into account. [18] developed robust EBLUP(REBLUP) in general linear mixed models, which they used Huber's $\phi$-function to modify certain "residual" terms by down-weighting contributions due to the outliers. By using a parametric bootstrap procedure, they have also developed estimators of the MSEs of the REBLUPs. Up to now, this method has been widely used in small area robust estimation. Later researches on robust SAE are completed on the basis of the above research. Such as, robust SAE in business surveys is discussed by using robust projection and M-quantile method in [22, 23] studied the robust estimation of nested error linear regression model by using huber's $\phi$-function and M-quantile based on hierarchical bayes theory and given prior information; The robust SAE of generalized linear models is discussed in [24, 25] reviewed the robust estimation of small area with outliers, and proposed Bootstrap MSE based on M-quantile estimators. [26] provide an overview of robust small area estimation. Readers who are interested in this research can refer to [22–27]. In this paper, we propose a new robust Bayes estimator using dendity power divergence, and investigate the proposed estimator's MSE and parameter estimation.

In the field of statistical inference, when outliers appear in the data, the density-based minimum distance is an effective method to solve this problem. The density power divergence (DPD) [4], which measures the discrepancy between two density functions, has been successfully used to build a robust estimator for independent and identically distributed observations. Since then, the method by minimzing DPD(MDPD) has been one of the most powerful tools in robust estimation. [28] extended the construction of the DPD and the corresponding minimum DPD estimator (MDPDE) to the case of independent but non-identically distributed data. The main idea of the DPD is to give small weight to the terms related to outliers, and then, the parameter estimation becomes robust against outliers. In the cases, the parameter $\alpha$ controls the trade-off between robustness and efficiency. The smaller the value of $\alpha$ is, the more efficient the model is; The higher the value of $\alpha$ is, the better the stability of the outlier is. The minimum divergence estimator corresponding to $\alpha = 0$ is the maximum likehood estimator(MLE). [29] used density power divergence to study robust Bayesian estimators, and discussed the asymptotic properties of the estimators and parameters. [30] discussed the theory of MDPD method in robust regression using the methods of S-estimation. This result showed that robust estimation based on MDPD method and Huber-$\phi$ function had similar effects. [31] employed the $\gamma$-divergence (similar to density power divergence) for the Fay-Herriot model and discussed empirical Bayes confidence intervals rather than MSE. Therefore, in this paper, we apply the MDPD method to the unit level model and compare it with the robust estimation proposed by [18].

In this paper, our main work is reflected in the following aspects. Firstly, the MDPD method is applied to the basic unit level model with outliers, and robust estimates of unknown parameters in the model are obtained. The asymptotic properties of the estimated parameters are further derived. Secondly, the selection algorithm of adjustment parameters in estimators is proposed to select the optimal tuning parameters. Thirdly, combined with parameter estimators, the general expression of robust small area estimator of unit level model is proposed. Fourthly, Bootstrap method is used to calculate the MSE of robust estimators, and the algorithm is given. Finally, the maximum likelihood estimator, the robust estimator in [18] and the proposed robust estimator are compared between simulated data and actual data to illustrate the efficiency and robustness properties of the estimators.

The rest of this paper is organized as follows. In Section 2, we introduce the basic unit level model and some notations used. In Section 3, the background and definition of MDPD

method are reviewed, and the robust estimation equations of model parameters are obtained by applying MDPD method to unit-level model. The asymptotic properties of robust parameters obtained by MDPD method are presented in Section 4. In Section 5, firstly, we propose robust empirical Bayes predicator based on MDPDE, secondly, we give an algorithm to select the optimum tuning parameter, and finally, we give the algorithm procedure to estimate MSE of robust EBP using Bootstrap method. In Section 6, we investigate performances of the proposed estimator by simulation and real data. The proposed method is compared with the robust method of [18] under different outlier generation backgrounds.

## 2 Basic unit level model

The NER model is a popular basic unit level model proposed by [5]. Suppose a finite survey population is partitioned into $m$ small areas, with the $i$-th area having $N_i$ units such that $\sum_{i=1}^{m} N_i = N$. We assume that $\mathbf{y}_{ij}$ is the value of a response variable $Y$ for the $j$ th unit in the $i$-th small area. The unit-specific auxiliary data $\mathbf{x}_{ij} = (x_{ij1}, \ldots, x_{ijp})^T$ are available for each population element $j(j = 1, \ldots, N_i)$ in each small area $i$. Then, the NER model described as

$$y_{ij} = \mathbf{x}_{ij}^T \boldsymbol{\beta} + v_i + e_{ij}, \quad j = 1, \ldots, N_i, \quad i = 1, \ldots m. \tag{1}$$

Where $\beta$ is a p-variate vector of unknown regression coefficients, the area-specific random effects $v_i$ are assumed to be independent $N(0, \sigma_v^2)$. The unit errors $e_{ij} = k_{ij}\tilde{e}_{ij}$ for known constants $k_{ij}$ and the $\tilde{e}_{ij}$'s are iid random variables independent of the $v_i$'s, with $N(0, \sigma_e^2)$.

We assume that a sample $s_i$ of size $n_i$ is taken from the $N_i$ units in the $i$ th area ($i = 1, \ldots, m$), and that the sample values also obey the assumed model (1). The latter assumption is satisfied under simple random sampling from each area or more generally for sampling designs that use the auxiliary information $\mathbf{x}_{ij}$ in the selection of the samples $s_i$. To see this, we write (1) in matrix form as

$$\mathbf{y}_i = \mathbf{x}_i \boldsymbol{\beta} + v_i \mathbf{1}_i + \mathbf{e}_i, \quad i = 1, \ldots, m, \tag{2}$$

where $\mathbf{x}_i$ is a $n_i \times p$ matrix, $\mathbf{y}_i$ and $\mathbf{e}_i$ are $n_i \times 1$ vectors, and $\mathbf{1}_i$ is the $n_i \times 1$ vector of ones.

Let $\boldsymbol{\theta} = \left( \boldsymbol{\beta}^T, \sigma_v^2, \sigma_e^2 \right)^T$ denote the unknown parameter of the model given in Eq (1). Under the assumption of normality of the model, $\mathbf{y}_i | \mathbf{x}_i$ obeys normal distribution. The conditional probability density is

$$f_{\boldsymbol{\theta}}\big(\mathbf{y}_i \mid \mathbf{x}_i\big) = (2\pi)^{-\frac{n_i}{2}} |\mathbf{V}_i|^{-\frac{1}{2}} \exp\left\{ -\frac{1}{2} (\mathbf{y}_i - \mathbf{x}_i \boldsymbol{\beta})^T \mathbf{V}_i^{-1} (\mathbf{y}_i - \mathbf{x}_i \boldsymbol{\beta}) \right\}, \tag{3}$$

where

$$\mathbf{V}_i = \sigma_e^2 \, \mathrm{diag}_{1 \le j \le n_i} (k_{ij}^2) + \sigma_v^2 \mathbf{1}_{n_i} \mathbf{1}_{n_i}^T.$$

The matrix $\mathbf{V}_i$ can be inverted explicitly. Using the Sherman-Morrison formula:

$$(\mathbf{A} + \mathbf{u}\mathbf{v}^T)^{-1} = \mathbf{A}^{-1} - \mathbf{A}^{-1}\mathbf{u}\mathbf{v}^T\mathbf{A}^{-1} / (1 + \mathbf{v}^T\mathbf{A}^{-1}\mathbf{u}),$$

and denoting

$$a_{ij} = k_{ij}^{-2}, \quad a_{i\cdot} = \sum_{j=1}^{n_i} a_{ij}, \quad \mathbf{a}_i = (a_{i1}, \ldots, a_{in_i})^T,$$

we get

$$\mathbf{V}_i^{-1} = \frac{1}{\sigma_e^2}\left[\operatorname{diag}_{1 \le j \le n_i}(a_{ij}) - \frac{\rho_i}{a_{i\cdot}}\mathbf{a}_i\mathbf{a}_i^T\right], |\mathbf{V}_i| = (\sigma_e^2 + a_{i\cdot}\sigma_v^2)\sigma_e^{2(n_i-1)}\prod_{j=1}^{n_i}k_{ij}^2. \tag{4}$$

where

$$\rho_i = \sigma_v^2/(\sigma_v^2 + \sigma_e^2/a_{i\cdot})$$

## 3 Density power divergence

### 3.1 Minimum density power divergence estimator

The density power divergence (DPD) measure was developed by [4] in terms of a tuning parameter $\gamma$. The DPD measure between the model density $f_\theta$ and the true density $g$ is defined as

$$d_\gamma(f_\theta, g) = \begin{cases} \int_y \left\{f_\theta^{1+\gamma}(y) - \left(1 + \frac{1}{\gamma}\right)f_\theta^\gamma(y)g(y) + \frac{1}{\gamma}g^{1+\gamma}(y)\right\} dy, & \text{for } \gamma > 0 \\ \int_y g(y)\log\left(\frac{g(y)}{f_\theta(y)}\right) dy, & \text{for } \gamma = 0 \end{cases}$$

where $\gamma$ is a tuning parameter. Note that, $G$ is not necessarily a member of the model family $F_\theta$. Further, for $\gamma = 0$, the DPD measure is obtained as a limiting case of $\gamma \to 0^+$, and is same as the Kullback-Leibler (KL) divergence. Generally, given a parametric model, we estimate $\theta$ by minimizing the DPD measure with respect to $\theta$ over its parametric space $\Theta$. We call the estimator the minimum power divergence estimator (MDPDE). It is well-known that, for $\gamma = 0$, minimization of the KL-divergent is equivalent to maximization of the log-likelihood function. Thus, the MLE can be considered as a special case of the MDPDE when $\gamma = 0$.

We substitute the conditional density $f_\theta(\mathbf{y}_i \mid \mathbf{X}_i)$ in 3 as the model density into the definition of DPD, and define the DPD measure based on SAE model as

$$d_\gamma(f_\theta, g) = \begin{cases} \int_\mathbf{x}\int_\mathbf{y}\left\{f_\theta^{1+\gamma}(\mathbf{y} \mid \mathbf{x}) - \left(1 + \frac{1}{\gamma}\right)f_\theta^\gamma(\mathbf{y} \mid \mathbf{x})g(\mathbf{y} \mid \mathbf{x}) + \frac{1}{\gamma}g^{1+\gamma}(\mathbf{y} \mid \mathbf{x})\right\} h(\mathbf{x})d\mathbf{x}d\mathbf{y}, & \text{for } \gamma > 0 \\ \int_\mathbf{x}\int_\mathbf{y}g(\mathbf{y} \mid \mathbf{x})\log\left(\frac{g(\mathbf{y}|\mathbf{x})}{f_\theta(\mathbf{y}|\mathbf{x})}\right) h(\mathbf{x})d\mathbf{x}d\mathbf{y}, & \text{for } \gamma = 0 \end{cases}$$

where $h(\mathbf{x})$ is the marginal probability density function of $\mathbf{X}$ and $g(\mathbf{y}|\mathbf{x})$ is the true conditional density of $\mathbf{Y}$ given $\mathbf{X}$ For $\gamma > 0$, after approximating the true distribution with the empirical, the DPD measure turns out to be

$$\widehat{d}_\gamma(f_\theta, g) = \frac{1}{m}\sum_{i=1}^m\int_\mathbf{y}f_\theta^{1+\gamma}(\mathbf{y}_i \mid \mathbf{x}_i)d\mathbf{y} - \frac{1+\gamma}{m\gamma}\sum_{i=1}^m f_\theta^\gamma(\mathbf{y}_i \mid \mathbf{x}_i) + c(\gamma). \tag{5}$$

where the last part of the expression in the right hand side of Eq (2) $c(\gamma) =$

$\frac{1}{m\gamma}\sum_{i=1}^m\int_\mathbf{y}g_i^{1+\gamma}(\mathbf{y} \mid \mathbf{x}_i)d\mathbf{y}$ is independent of the unknown parameter $\theta = \left(\boldsymbol{\beta}^T, \sigma_v^2, \sigma_e^2\right)^T$. Hence, Eq (5) simplifies to

$$\widehat{d}_\gamma(f_\theta, g) = \sum_{i=1}^m(2\pi)^{-\frac{n_i\gamma}{2}}|\mathbf{V}_i|^{-\frac{\gamma}{2}}\left[(1+\gamma)^{-\frac{n_i}{2}} - \frac{1+\gamma}{\gamma}\exp\left[-\frac{\gamma}{2}B_i\right]\right] + c(\gamma). \tag{6}$$

where $B_i = (\mathbf{y}_i - \mathbf{x}_i\beta)^T \mathbf{V}_i^{-1}(\mathbf{y}_i - \mathbf{x}_i\beta)$. Using the formula (4), we get

$$B_i = \frac{1}{\sigma_e^2}\sum_{j=1}^{n_i} a_{ij}(y_{ij} - \mathbf{x}_{ij}\boldsymbol{\beta})^2 - \frac{\rho_i}{\sigma_e^2 a_{i\cdot}}\left\{\sum_{j=1}^{n_i} a_{ij}(y_{ij} - \mathbf{x}_{ij}\boldsymbol{\beta})\right\}^2$$

The MDPDE of $\boldsymbol{\theta}$ is obtained by minimizing $\widehat{d}_\gamma(f_\theta, g)$ over $\boldsymbol{\theta} \in \boldsymbol{\Theta}$, where $\boldsymbol{\Theta}$ is the parameter space composed by all possible parameters $\boldsymbol{\theta}$. Obviously, if the $i$-th observation is an outlier, then the value of the conditional density $f_\theta(\mathbf{y}_i \mid \mathbf{x}_i)$ is smaller compared to other observations. In this way, the second term of Eq (5) is negligible when $\gamma > 0$, thus the corresponding MDPDE becomes robust against outlier. The tuning parameter $\gamma$ controls the trade-off between efficiency and robustness of MDPDE. When $\gamma$ increases, robustness increases and efficiency decreases, and vice versa. In addition, when $\gamma = 0$, the DPD becomes KL divergence, and MDPDE becomes MLE. At this time, for an outlying observation, the KL divergence measure diverges as $f_\theta(\mathbf{y}_i \mid \mathbf{x}_i) \to 0$, and MLE method is invalid.

The partial derivative of $\widehat{d}_\gamma(f_\theta, g)$ with respect to $\boldsymbol{\theta} = \left(\boldsymbol{\beta}^T, \sigma_v^2, \sigma_e^2\right)^T$ in Eq (5) is taken to obtain the following estimated equation:

$$\sum_{i=1}^{m}(2\pi)^{-\frac{n_i\gamma}{2}}|\mathbf{V}_i|^{-\frac{\gamma}{2}}\quad \exp(-\frac{\gamma}{2}B_i)\sum_{j=1}^{n_i}a_{ij}(y_{ij} - \mathbf{x}_{ij}\boldsymbol{\beta})\mathbf{x}_{ij}^T$$
$$= \sum_{i=1}^{m}(2\pi)^{-\frac{n_i\gamma}{2}}|\mathbf{V}_i|^{-\frac{\gamma}{2}}\exp(-\frac{\gamma}{2}B_i)\rho_i\bar{\mathbf{x}}_i^T\sum_{j=1}^{n_i}a_{ij}(y_{ij} - \mathbf{x}_{ij}\boldsymbol{\beta}),$$

(7)

$$\sum_{i=1}^{m}(2\pi)^{-\frac{n_i\gamma}{2}}|\mathbf{V}_i|^{-\frac{\gamma}{2}-1}\qquad\left(\gamma(1+\gamma)^{-\frac{n_i+2}{2}} - \exp(-\frac{\gamma}{2}B_i)\right)a_{i\cdot}\sigma_e^{2(n_i-1)}\prod_{j=1}^{n_i}k_{ij}^2$$
$$= -\sum_{i=1}^{m}(2\pi)^{-\frac{n_i\gamma}{2}}\quad|\mathbf{V}_i|^{-\frac{\gamma}{2}}\exp(-\frac{\gamma}{2}B_i)\frac{1}{(a_{i\cdot}\sigma_v^2 + \sigma_e^2)^2}\left(\sum_{j=1}^{n_i}a_{ij}(y_{ij} - \mathbf{x}_{ij}\boldsymbol{\beta})\right)^2$$

(8)

$$\sum_{i=1}^{m}(2\pi)^{-\frac{n_i\gamma}{2}}|\mathbf{V}_i|^{-\frac{\gamma}{2}-1}\left(\gamma(1+\gamma)^{-\frac{n_i+2}{2}} - \exp(-\frac{\gamma}{2}B_i)\right)\left(n_i\sigma_e^{2(n_i-1)} + (n_i - 1)a_{i\cdot}\sigma_v^2\sigma_e^{2(n_i-2)}\right)\prod_{j=1}^{n_i}k_{ij}^2$$
$$= \sum_{i=1}^{m}(2\pi)^{-\frac{n_i\gamma}{2}}|\mathbf{V}_i|^{-\frac{\gamma}{2}}\exp(-\frac{\gamma}{2}B_i)\left\{ -\frac{1}{\sigma_e^4}\sum_{j=1}^{n_i}a_{ij}(y_{ij} - \mathbf{x}_{ij}\boldsymbol{\beta})^2 \right.$$
$$\left. + \frac{2\sigma_e^2\sigma_v^2 + a_{i\cdot}\sigma_v^4}{(a_{i\cdot}\sigma_v^2 + \sigma_e^2)^2\sigma_e^4}\left(\sum_{j=1}^{n_i}a_{ij}(y_{ij} - \mathbf{x}_{ij}\boldsymbol{\beta})\right)^2\right\}$$

(9)

## 3.2 Choice of the optimal tuning parameter $\gamma$

It can be seen from the above results that the unknown tuning parameter $\gamma$ is included in the estimated expression of parameter $\theta$ which is iterated by MDPD method. The choice of $\gamma$ determines the trade-off between robustness and statistical efficiency [4]. When the value of $\gamma$ is closer to 1, the estimated parameter has stronger robustness; otherwise, the weaker the robustness, the stronger the efficiency. Therefore, choosing the appropriate tuning parameter $\gamma$ is the key factor in robust estimation. we hope to choose a data-driven value of $\gamma$ in an

optimal way which balances the concerns of robustness and efficiency. At present, there are two main methods for selecting tuning parameters. One is based on the proportion relationship between efficiency and robustness. Researchers determine the proportion between them according to their own needs, and then select the optimal tuning parameters. For example, [29] selects tuning parameters when using DPD method for area level estimation. Another method is based on data-driven parameter selection method, [32] minimizes the MSE of the estimated parameter to get the optimal tuning parameter, but this method depends on the selection of initial value, different pilot value of parameter may result in different tuning parameters. [33] Based on [32], a method of adjusting parameter selection that does not depend on the initial value is proposed. In this paper, we will use the parameter selection method mentioned in [33] to select the optimal tuning parameters for constructing robust small area estimators.

For the true value $\theta^*$ of the unknown parameter $\theta$, the optimal tuning parameter $\gamma$ is obtained by minimizing the summed MSE of the MDPD estimator, i.e

$$E\left\{ \left(\widehat{\boldsymbol{\theta}}_{\gamma} - \boldsymbol{\theta}^*\right)^{\mathrm{T}} \left(\widehat{\boldsymbol{\theta}}_{\gamma} - \boldsymbol{\theta}^*\right) \right\} = N^{-1} \operatorname{tr} \{J_{\gamma}^{-1}(\boldsymbol{\theta}_{\gamma}) K_{\gamma}(\boldsymbol{\theta}_{\gamma}) J_{\gamma}^{-1}(\boldsymbol{\theta}_{\gamma})\} + (\boldsymbol{\theta}_{\gamma} - \boldsymbol{\theta}^*)^{\mathrm{T}} (\boldsymbol{\theta}_{\gamma} - \boldsymbol{\theta}^*), \quad (10)$$

As the unknown parameter $\theta^*$ is contained in the formula (10), there are usually two ways to select the optimal tuning parameter $\gamma$. One method is to think that the estimated $\widehat{\theta}_{\gamma}$ is the true value of parameter $\theta^*$, that is, the optimal tuning parameter can be obtained only by solving the minimum value of the first term of the above equation. This method is easy to use, but we know that $\widehat{\theta}_{\gamma}$ is asymptotically tend to $\theta$, so direct substitution will produce a large error. Another method is to set an initial value $\theta_p$ of $\theta^*$ and then minimize (10) to find the optimal parameter. This method is used in [32], and wick-Jones (WJ) algorithm is given to select the optimal tuning parameters. However, the WJ algorithm relies heavily on the selection of the initial $\theta^p$, which directly determines the selection of the optimal parameter. In order to overcome the shortcomings of the above two selection methods, [33] proposed iterative WJ (IWJ) algorithm, which is used to calculate the optimal tuning parameters. In this paper, this method is also used to select the optimal tuning parameters. The specific algorithm steps are as follows:

**Algorithm 1** IWJ algorithm

**Input:** set the initial $\gamma$, $\boldsymbol{\theta}^{(0)} = (\boldsymbol{\beta}^{(0)}, \sigma_e^{2(0)}, \sigma_v^{2(0)})$
**Repeat:**
1: WJ algorithm is used to minimize Formula (10) and update $\gamma$ within interval $I_{\gamma}$,

$$\gamma^{(i+1)} = min_{\gamma \in I_{\gamma}} E\left\{ \left(\widehat{\boldsymbol{\theta}}_{\gamma} - \boldsymbol{\theta}^{*(i)}\right)^{\mathrm{T}} \left(\widehat{\boldsymbol{\theta}}_{\gamma} - \boldsymbol{\theta}^{*(i)}\right) \right\}.$$

2: Fix $\gamma^{(i+1)}$, put it into the MDPD iteration program, get the estimate of $\widehat{\boldsymbol{\theta}}$, and update it $\boldsymbol{\theta}^{*(i+1)} \leftarrow$ MDPD($\boldsymbol{\theta}^{*(i+1)}$).
3: repeat step 1,2, until $|\gamma^{(i+1)} - \gamma^{(i)}| < \epsilon$ or $|\boldsymbol{\theta}^{*(i+1)} - \boldsymbol{\theta}^{*(i)}| < \epsilon^*$, where $\epsilon$ and $\epsilon^*$ is the accuracy of parameter estimation.
**Output:** $\boldsymbol{\theta}^{(m+1)}$

## 4 Asymptotic distribution of the robust estimator

In this section, we investigate the asymptotic distribution of the robust estimator of model parameters, when the data generating distribution $G(\mathbf{y} \mid \mathbf{x})$ is not necessarily in the model

famliy. Let's define the score function as $\mathbf{u}_\theta(\mathbf{y}_i \mid \mathbf{x}_i) = \frac{\partial}{\partial\theta}\log f_\theta(\mathbf{y}_i \mid \mathbf{x}_i)$

$$\mathbf{u}_\theta(\mathbf{y}_i \mid \mathbf{x}_i) = \left(\mathbf{u}_\beta^T(\mathbf{y}_i \mid \mathbf{x}_i), \mathbf{u}_{\sigma_v^2}^T(\mathbf{y}_i \mid \mathbf{x}_i), \mathbf{u}_{\sigma_e^2}^T(\mathbf{y}_i \mid \mathbf{x}_i)\right)^T. \tag{11}$$

According to the definition of score function and (3), we can get

$$\mathbf{u}_\beta(\mathbf{y}_i \mid \mathbf{x}_i) = \frac{\partial}{\partial\beta}\log f_\theta(\mathbf{y}_i \mid \mathbf{x}_i) = \frac{1}{\sigma_e^2}\sum_{j=1}^{n_i}a_{ij}x_{ij}(y_{ij} - x_{ij}\beta) - \frac{a_{i\cdot}\sigma_v^2\overline{\mathbf{x}}_i}{\sigma_e^2(\sigma_e^2 + a_{i\cdot}\sigma_v^2)}\sum_{j=1}^{n_i}a_{ij}(y_{ij} - x_{ij}\beta),$$

$$\mathbf{u}_{\sigma_v^2}(\mathbf{y}_i \mid \mathbf{x}_i) = \frac{\partial}{\partial\sigma_v^2}\log f_\theta(\mathbf{y}_i \mid \mathbf{x}_i) = -\frac{a_{i\cdot}}{2(\sigma_e^2 + a_{i\cdot}\sigma_v^2)} + \frac{1}{2(\sigma_e^2 + a_{i\cdot}\sigma_v^2)^2}\left\{\sum_{j=1}^{n_i}a_{ij}(y_{ij} - x_{ij}\beta)\right\}^2,$$

$$\mathbf{u}_{\sigma_e^2}(\mathbf{y}_i \mid \mathbf{x}_i) = \frac{\partial}{\partial\sigma_e^2}\log f_\theta(\mathbf{y}_i \mid \mathbf{x}_i) = -\frac{n_i\sigma_e^2 + (n_i - 1)a_{i\cdot}\sigma_v^2}{2\sigma_e^2(\sigma_e^2 + a_{i\cdot}\sigma_v^2)} \tag{12}$$

$$+ \frac{1}{2\sigma_e^4}\sum_{j=1}^{n_i}a_{ij}(y_{ij} - x_{ij}\beta)^2 - \frac{\sigma_v^2(2\sigma_e^2 + a_{i\cdot}\sigma_v^2)}{2\sigma_e^4(\sigma_e^2 + a_{i\cdot}\sigma_v^2)^2}\left\{\sum_{j=1}^{n_i}a_{ij}(y_{it} - x_{it}\beta)\right\}^2.$$

For $i = 1, 2, \cdots, N$, we define

$$\mathbf{J}^{(i)} = \int_\mathbf{y} \mathbf{u}_\theta(\mathbf{y} \mid \mathbf{x}_i)\mathbf{u}_\theta^T(\mathbf{y} \mid \mathbf{x}_i)f_\theta^{1+\gamma}(\mathbf{y} \mid \mathbf{x}_i)d\mathbf{y}$$

$$+ \int_\mathbf{y}\{I_\theta(\mathbf{y} \mid \mathbf{x}_i) - \gamma\mathbf{u}_\theta(\mathbf{y} \mid \mathbf{x}_i)\mathbf{u}_\theta^T(\mathbf{y} \mid \mathbf{x}_i)\}\{g(\mathbf{y} \mid \mathbf{x}_i) - f_\theta(\mathbf{y} \mid \mathbf{x}_i)\}f_\theta^\gamma(\mathbf{y} \mid \mathbf{x}_i)d\mathbf{y},$$

$$\mathbf{K}^{(i)} = \int_\mathbf{y} \mathbf{u}_\theta(\mathbf{y} \mid \mathbf{x}_i)\mathbf{u}_\theta^T(\mathbf{y} \mid \mathbf{x}_i)f_\theta^{2\gamma}(\mathbf{y} \mid \mathbf{x}_i)g(\mathbf{y} \mid \mathbf{x}_i)d\mathbf{y} - \xi^{(i)}\xi^{(i)T},$$

$$\mathbf{I}_\theta(\mathbf{y} \mid \mathbf{x}_i) = -\frac{\partial}{\partial\theta}\mathbf{u}_\theta(\mathbf{y} \mid \mathbf{x}_i), \quad \xi^{(i)} = \int_\mathbf{y} u_\theta(\mathbf{y} \mid \mathbf{x}_i)f_\theta^\gamma(\mathbf{y} \mid \mathbf{x}_i)g(\mathbf{y} \mid \mathbf{x}_i)d\mathbf{y}.$$

We further define $\mathbf{J} = \lim_{m\to\infty}\frac{1}{m}\sum_{i=1}^m\mathbf{J}^{(i)}$, $\mathbf{K} = \lim_{m\to\infty}\frac{1}{m}\sum_{i=1}^m\mathbf{K}^{(i)}$. For the asymptotic distribution of the MDPDE, we need the following assumptions:

1. The true density $g(y \mid x)$ is supported over the entire real line $\mathbb{R}$;

2. There is an open subset $\omega \in \Theta_0$ containing the best fitting parameter $\theta$ such tat $\mathbf{J}$ is positive definite for all $\theta \in \omega$;

3. There exist functions $M_{jkl}(x, y)$ such that $|\partial^3 \exp\left[(\mathbf{y} - \mathbf{x}\beta)^T V_i^{-1}(\mathbf{y} - \mathbf{x}\beta)\right]/\partial\theta_j\partial\theta_k\partial\theta_l| \leq M_{jkl}(x, y)$ for all $\theta \in \omega$, where $\int_x\int_y|M_{jkl}(x, y)|g(y \mid x)h(x)dydx < \infty$ for all $j$, $k$ and $l$.

**Theorem 4.1** *Under the regularity conditions (1)-(3), with probability tending to 1 as $m \to \infty$, there exists $\widehat{\theta}$, such that*

1. *$\widehat{\theta}$ is consistent for $\theta$;*

2. *the asymptotic distribution of $\widehat{\theta}$ is given by*

$$\sqrt{m}(\widehat{\theta} - \theta) \sim N_{p+2}(0, \mathbf{J}^{-1}\mathbf{K}\mathbf{J}^{-1}).$$

**Proof**: The proof of the theorem is given in S1 File.

Note that, if the true distribution $g(\mathbf{y} \mid \mathbf{x})$ is a member of the model family $f_\theta(\mathbf{y}|\mathbf{x})$ for some $\boldsymbol{\theta} \in \boldsymbol{\Theta}$, then

$$\mathbf{J}^{(i)} = \int_{\mathbf{y}} \mathbf{u}_{\boldsymbol{\theta}}(\mathbf{y} \mid \mathbf{x}_i)\mathbf{u}_{\boldsymbol{\theta}}^T(\mathbf{y} \mid \mathbf{x}_i)f_{\boldsymbol{\theta}}^{1+\gamma}(\mathbf{y} \mid \mathbf{x}_i)d\mathbf{y},$$

$$\mathbf{K}^{(i)} = \int_{\mathbf{y}} \mathbf{u}_{\boldsymbol{\theta}}(\mathbf{y} \mid \mathbf{x}_i)\mathbf{u}_{\boldsymbol{\theta}}^T(\mathbf{y} \mid \mathbf{x}_i)f_{\boldsymbol{\theta}}^{2\gamma+1}(\mathbf{y} \mid \mathbf{x}_i)d\mathbf{y} - \boldsymbol{\xi}^{(i)}\boldsymbol{\xi}^{(i)T}, \qquad (13)$$

$$\boldsymbol{\xi}^{(i)} = \int_{\mathbf{y}} u_{\boldsymbol{\theta}}(\mathbf{y} \mid \mathbf{x}_i)f_{\boldsymbol{\theta}}^{\gamma+1}(\mathbf{y} \mid \mathbf{x}_i)d\mathbf{y}.$$

In this case, the symmetric matrix $\mathbf{J}^{(i)}$ can be partitioned as

$$\mathbf{J}^{(i)} = \begin{bmatrix} \mathbf{J}_{\boldsymbol{\beta}}^{(i)} & \mathbf{J}_{\boldsymbol{\beta},\sigma_v^2}^{(i)} & \mathbf{J}_{\boldsymbol{\beta},\sigma_e^2}^{(i)} \\ \cdot & \mathbf{J}_{\sigma_v^2}^{(i)} & \mathbf{J}_{\sigma_v^2,\sigma_e^2}^{(i)} \\ \cdot & \cdot & \mathbf{J}_{\sigma_e^2}^{(i)} \end{bmatrix}.$$

Combining Eqs (12) and (13), the elements in $\mathbf{J}^{(i)}$ can be deduced. Detailed calculation and results can be found in S1 File.

Similarly, $\boldsymbol{\xi}^{(i)}$ can be partitioned as $\boldsymbol{\xi}^{(i)} = (\xi_{\beta}^{(i)T}, \xi_{\sigma_v^2}^{(i)}, \xi_{\sigma_e^2}^{(i)})^T$, In S1 File, the derivation formula of the components of $\boldsymbol{\xi}^{(i)}$ is given.

Note that if we write the matrix $\mathbf{J}^{(i)}$ as a function of $\gamma$, i.e. $\mathbf{J}^{(i)} \equiv \mathbf{J}^{(i)}(\gamma)$, Based on the representation of $\mathbf{K}^{(i)}$ and $\mathbf{J}^{(i)}$ in (13), we have

$$\mathbf{K}^{(i)} = \mathbf{J}^{(i)}(2\gamma) - \boldsymbol{\xi}^{(i)}\boldsymbol{\xi}^{(i)T}.$$

Therefore, $\mathbf{K}$ can be written as

$$\mathbf{K} = \lim_{N \to \infty} \frac{1}{N} \sum_{i=1}^{N} \mathbf{J}^{(i)}(2\gamma) - \boldsymbol{\xi}^{(i)}\boldsymbol{\xi}^{(i)T}.$$

Through the calculation of the above covariance matrix, it can be seen that the parameter variance increases with the increase of $\gamma$, which indicates that the efficiency of MDPDE decreases with the increase of $\gamma$. This further verifies that the tuning parameter $\gamma$ is used to control the trade-off between efficiency and robustness of MDPDE, and that robustness increases and efficiency decreases as $\gamma$ increases However, our subsequent simulations show that this loss of efficiency is not significant.

# 5 Robust empirical Bayes perdictor and MSE

## 5.1 Robust EB predictor under a finite population

In this section, we discuss EB estimators of parameters of a finite population. A finite population $P$ contains $N$ units and a sample $s$ of size $n$ is drawn from $P$. We denote by $\mathbf{y}^P$ the unit values vector of the target variable in the population, which is assumed to be random with a given joint probability distribution. We write $\mathbf{y}_s$ as the subvector of $\mathbf{y}^P$ composed of sampling units, $\mathbf{y}_r$ as the subvector composed of unsampled units and assume without loss of generality that the first $n$ units of $\mathbf{y}$ are the sample elements, that is, $\mathbf{y}^P = (\mathbf{y}_s^T, \mathbf{y}_r^T)^T$.

We assume that the vaule $y_{ij}$ of a target variable for $j$th unit in $i$th area follows the basic unit level model (1). At the same time, we assume that $\mathbf{y}_i$ obeys normal distribution under the condition of auxiliary information $\mathbf{X}_i$, i.e $\mathbf{y}_i \sim N(\mathbf{X}_i\boldsymbol{\beta}, \mathbf{V}_i)$. We next partition (2) into sampled and

nonsampled parts:

$$\mathbf{y}_i = \begin{bmatrix} \mathbf{y}_{is} \\ \mathbf{y}_{ir} \end{bmatrix} = \begin{bmatrix} \mathbf{X}_{is} \\ \mathbf{X}_{ir} \end{bmatrix} \boldsymbol{\beta} + v_i \begin{bmatrix} \mathbf{1}_{is} \\ \mathbf{1}_{ir} \end{bmatrix} + \begin{bmatrix} \mathbf{e}_{is} \\ \mathbf{e}_{ir} \end{bmatrix},$$

where the subscript $r$ denotes the nonsampled units. The covariance matrix can be decomposited as:

$$\mathbf{V}_i = \begin{pmatrix} \mathbf{V}_{is} & \mathbf{V}_{isr} \\ \mathbf{V}_{irs} & \mathbf{V}_{ir} \end{pmatrix},$$

where

$$\mathbf{V}_{is} = \mathbf{R}_{is} + \sigma_v^2 \mathbf{1}_{n_i} \mathbf{1}_{n_i}^T = \text{diag}_{1 \le j \le n_i} (k_{ij}^2) \sigma_e^2 + \sigma_v^2 \mathbf{1}_{n_i} \mathbf{1}_{n_i}^T,$$
$$\mathbf{V}_{ir} = \mathbf{R}_{ir} + \sigma_v^2 \mathbf{1}_{N_i - n_i} \mathbf{1}_{N_i - n_i}^T = \text{diag}_{n_i < j < N_i} (k_{ij}^2) \sigma_e^2 + \sigma_v^2 \mathbf{1}_{N_i - n_i} \mathbf{1}_{N_i - n_i}^T,$$
$$\mathbf{V}_{isr} = \sigma_v^2 \mathbf{1}_{n_i} \mathbf{1}_{N_i - n_i}^T, \quad \mathbf{V}_{irs} = \sigma_v^2 \mathbf{1}_{N_i - n_i} \mathbf{1}_{n_i}^T$$

The non-sampled sub-vectors $\boldsymbol{y}_{ir}$ follow the marginal models derived from the population model (1), i.e.

$$\boldsymbol{y}_{ir} = \boldsymbol{X}_{ir} \boldsymbol{\beta} + \mathbf{1}_{N_i - n_i} v_i + \boldsymbol{e}_{ir}, \quad i = 1, \ldots, m. \tag{14}$$

The vectors $\boldsymbol{e}_{ir}$ are independent with $\boldsymbol{e}_{ir} \sim N(\mathbf{0}_{N_i - n_i}, \sigma_e^2 \boldsymbol{W}_{ir})$, where $\boldsymbol{W}_{ir} = \text{diag}_{n_i < j < N_i} (k_{ij}^2)$. The vectors $\boldsymbol{y}_{ir}$ are independent and normally distributed with

$$\boldsymbol{y}_{ir} \sim N(\boldsymbol{\mu}_{ir}, \boldsymbol{V}_{ir}),$$

where $\boldsymbol{\mu}_{ir} = \boldsymbol{X}_{ir} \boldsymbol{\beta}$.

The distribution of $\boldsymbol{y}_{ir}$, given the sample data $\boldsymbol{y}_{is}$, is

$$\boldsymbol{y}_{ir} | \boldsymbol{y}_s \sim \boldsymbol{y}_{ir} | \boldsymbol{y}_{is} \sim N\left( \boldsymbol{\mu}_{ir|s}, \boldsymbol{V}_{ir|s} \right). \tag{15}$$

The conditional mean vector is

$$\boldsymbol{\mu}_{ir|s} = \boldsymbol{\mu}_{ir} + \boldsymbol{V}_{irs} \boldsymbol{V}_{is}^{-1} (\boldsymbol{y}_{is} - \boldsymbol{\mu}_{is}) = \boldsymbol{X}_{ir} \boldsymbol{\beta} + \sigma_v^2 \mathbf{1}_{N_i - n_i} \mathbf{1}'_{n_i} \boldsymbol{V}_{is}^{-1} (\boldsymbol{y}_{is} - \boldsymbol{X}_{is} \boldsymbol{\beta}) \tag{16}$$

and the conditional covariance matrix is

$$\begin{aligned}
\boldsymbol{V}_{ir|s} &= \boldsymbol{V}_{ir} - \boldsymbol{V}_{irs} \boldsymbol{V}_{is}^{-1} \boldsymbol{V}_{isr} = \sigma_v^2 \mathbf{1}_{N_i - n_i} \mathbf{1}_{N_i - n_i}^T + \sigma_e^2 \boldsymbol{W}_{ir} \\
&\quad - \sigma_v^2 \mathbf{1}_{N_i - n_i} \mathbf{1}_{n_i}^T \frac{1}{\sigma_e^2} \left( \boldsymbol{W}_{ir}^{-1} - \frac{\sigma_v^2}{\sigma_e^2 + a_{ir\cdot} \sigma_v^2} \boldsymbol{a}_{ir} \boldsymbol{a}_{ir}^T \right) \sigma_v^2 \mathbf{1}_{n_i} \mathbf{1}_{N_i^T - n_i} \\
&= \sigma_v^2 \mathbf{1}_{N_i - n_i} \mathbf{1}_{N_i - n_i}^T + \sigma_e^2 \boldsymbol{R}_{ir} - \frac{\sigma_v^4}{\sigma_e^2} \left( a_{ir\cdot} \mathbf{1}_{N_i - n_i} \mathbf{1}_{N_i - n_i}^T - \frac{\sigma_v^2}{\sigma_e^2 + a_{ir\cdot}} a_{ir\cdot}^2 \mathbf{1}_{N_i - n_i} \mathbf{1}_{N_i - n_i}^T \right) \\
&= \sigma_v^2 \left( 1 - \frac{\sigma_v^2}{\sigma_e^2} a_{ir\cdot} (1 - \rho_{ir}) \right) \mathbf{1}_{N_i - n_i} \mathbf{1}'_{N_i - n_i} + \sigma_e^2 \boldsymbol{R}_{ir} \\
&= \sigma_v^2 (1 - \rho_{ir}) \mathbf{1}_{N_i - n_i} \mathbf{1}'_{N_i - n_i} + \sigma_e^2 \boldsymbol{R}_{ir}
\end{aligned} \tag{17}$$

where $\boldsymbol{a}_{ir} = (a_{in_i + 1}, \ldots, a_{iN_i})^T$, $a_{ir\cdot} = \sum_{j=n_i+1}^{N_i} a_{ij}$, $\rho_{ir} = \frac{a_{ir\cdot} \sigma_v^2}{\sigma_e^2 + a_{ir\cdot} \sigma_v^2}$

If $n_i \neq 0$ and $j \in U_i - s_i$, the conditional mean is

$$
\begin{aligned}
\mu_{ij|s} &= x_{ij}\boldsymbol{\beta} + \frac{\sigma_v^2}{\sigma_e^2}\mathbf{1}_{n_i}^T\left(\mathbf{W}_{is}^{-1} - \frac{\sigma_v^2}{\sigma_e^2 + a_{is\cdot}\sigma_v^2}\boldsymbol{a}_{is}\boldsymbol{a}_{is}^T\right)(\boldsymbol{y}_{is} - \boldsymbol{X}_{is}\boldsymbol{\beta}) \\
&= x_{ij}\boldsymbol{\beta} + \frac{\sigma_v^2}{\sigma_e^2}\left(1 - \frac{a_{is\cdot}\sigma_v^2}{\sigma_e^2 + a_{is\cdot}\sigma_v^2}\right)\boldsymbol{a}_{is}^T(\boldsymbol{y}_{is} - \boldsymbol{X}_{is}\boldsymbol{\beta}) \\
&= x_{ij}\boldsymbol{\beta} + \rho_{is}(\overline{\boldsymbol{Y}}_{is}^w - \overline{\boldsymbol{X}}_{is}^w\boldsymbol{\beta})
\end{aligned}
$$

where $\overline{\boldsymbol{Y}}_{is}^w = a_{is\cdot}^{-1}\boldsymbol{a}_{is}^T\boldsymbol{y}_{is} = a_{is\cdot}^{-1}\sum_{j=1}^{n_i}a_{ij}y_{ij}$ and $\overline{\boldsymbol{X}}_i^w = a_{is\cdot}^{-1}\boldsymbol{a}_{is}^T\boldsymbol{X}_{is} = a_{is\cdot}^{-1}\sum_{j=1}^{n_i}a_{ij}\boldsymbol{x}_{ij}$. For any $j \in U_i - s_i$, it thus holds that

$$
\mu_{ij|s} = \begin{cases} x_{ij}\beta + \rho_{is}(\overline{\boldsymbol{Y}}_{is}^w - \overline{\boldsymbol{X}}_{is}^w\boldsymbol{\beta}) & \text{if } n_i \neq 0, \\ x_{ij}\beta & \text{if } n_i = 0, \end{cases} \qquad \rho_{is} = \frac{a_{is\cdot}\sigma_v^2}{a_{is\cdot}\sigma_v^2 + \sigma_e^2}
$$

For any $j \in U_i - s_i$, the conditional variance is

$$
v_{ij|s} = \begin{cases} \sigma_v^2(1 - \rho_{is}) + k_{ij}^2\sigma_e^2 & \text{if } n_i \neq 0 \\ \sigma_v^2 + k_{ij}^2\sigma_e^2 & \text{if } n_d = 0 \end{cases}
$$

In general, our goal is to use the available sample data $\mathbf{y}_s$ to estimate the value of the real measurable function $\tau = h(\mathbf{y}^p)$ with respect to the population vector $\mathbf{y}^p$. Therefore, The conditional distribution of $\mathbf{y}_r$, given $\mathbf{y}_s$, plays an important role in the calculations of the best predictors (BPs) of population parameters $\tau = h(\mathbf{y})$. Assume that the model parameters $\boldsymbol{\theta} = \left(\boldsymbol{\beta}^T, \sigma_v^2, \sigma_e^2\right)^T$ are known. Under the unit level model, the BP is an unbiased predictor $\widehat{\tau} = \widehat{h}(\boldsymbol{y}_s)$ of $\tau$ that minimizes the MSE. According to [8], the BP of $\tau$ is $\widehat{\tau}^{bp} = E_{\boldsymbol{y}_r}[\tau \mid \boldsymbol{y}_s]$.

Therefore, the EB estimator of $\tau$ can be obtained by using formula (15)–(17), $\widehat{\tau} = \widehat{h}(\boldsymbol{y}^P) = h(\mathbf{y}_s, \widehat{\mathbf{y}}_r)$, where $\widehat{\mathbf{y}}_r = \boldsymbol{\mu}_{ir|s} + \widehat{\mathbf{v}}_{ir|s}$, and $\widehat{\mathbf{v}}_{ir|s} \sim N(\mathbf{0}, \boldsymbol{V}_{ir|s})$. In practice, the model parameters $\boldsymbol{\theta} = \left(\boldsymbol{\beta}^T, \sigma_v^2, \sigma_e^2\right)^T$ are replaced by consistent estimates $\widehat{\boldsymbol{\theta}} = \left(\widehat{\boldsymbol{\beta}}^T, \widehat{\sigma}_v^2, \widehat{\sigma}_e^2\right)^T$, and then the variables $\widehat{\mathbf{y}}_r$ are generated from (15), thus, the EBLUP of $\widehat{\tau} = \widehat{h}(\mathbf{y}^P)$ can be obtained.

## 5.2 EBLUP of area means

In this section, we derives the EBLUPs of $\overline{Y}_i$, where $\overline{Y}_i = \frac{1}{N_i}\sum_{j=1}^{N_i}y_{ij}$. Let $\widehat{\boldsymbol{\beta}}$, $\widehat{\sigma}_v^2$, and $\widehat{\sigma}_e^2$ be consistent estimators of the model parameters $\boldsymbol{\beta}$, $\sigma_v^2$, and $\sigma_e^2$, respectively. Under the conditioned distribution (15), the predicted values are

$$
\widehat{\boldsymbol{y}}_{is}^{ebp} = \boldsymbol{y}_{is}, \quad \widehat{\boldsymbol{y}}_{ir}^{ebp} = \widehat{\boldsymbol{\mu}}_{ir|s} = \boldsymbol{X}_{ir}\widehat{\boldsymbol{\beta}} + \widehat{\sigma}_v^2\mathbf{1}_{N_i-n_i}\mathbf{1}_{n_i}'\widehat{\boldsymbol{V}}_{is}^{-1}(\boldsymbol{y}_{is} - \boldsymbol{X}_{is}\widehat{\boldsymbol{\beta}}). \tag{18}
$$

or equivalently

$$
\widehat{y}_{ij}^{ebp} = \begin{cases} y_{ij} & \text{if } j \in s_i. \\ x_{ij}\widehat{\boldsymbol{\beta}} + \rho_{is}(\overline{\boldsymbol{Y}}_{is}^w - \overline{\boldsymbol{X}}_{is}^w\boldsymbol{\beta}) & \text{if } j \in r_i = U_i - s_i. \end{cases}
$$

The EBLUP of $\overline{Y}_d$ is

$$
\begin{aligned}
\widehat{\overline{Y}}_i^{ebp} &= \frac{1}{N_i} \sum_{j=1}^{N_i} \widehat{y}_{ij}^{ebp} = \frac{1}{N_i} \sum_{j \in s_i} y_{ij} + \frac{1}{N_i} \sum_{j \in r_i} \left\{ \boldsymbol{x}_{ij} \widehat{\boldsymbol{\beta}} + \widehat{v}_i \right\} \\
&= f_i \widehat{\overline{Y}}_i + \frac{1}{N_i} \sum_{j \in U_i} \left\{ \boldsymbol{x}_{ij} \widehat{\boldsymbol{\beta}} + \widehat{u}_i \right\} - f_i \frac{1}{n_i} \sum_{j \in s_i} \left\{ \boldsymbol{x}_{ij} \widehat{\boldsymbol{\beta}} + \widehat{u}_i \right\} \\
&= (1 - f_i) [\overline{\boldsymbol{X}}_i \widehat{\boldsymbol{\beta}} + \widehat{u}_i] + f_i [\widehat{\overline{Y}}_i + (\overline{\boldsymbol{X}}_i - \widehat{\overline{\boldsymbol{X}}}_i) \widehat{\boldsymbol{\beta}}]
\end{aligned} \tag{19}
$$

where $f_i = \frac{n_i}{N_i}$ is the domain sample fraction, $\widehat{\overline{Y}}_i = \frac{1}{n_i} \sum_{j \in s_i} y_{ij}$, $\widehat{\overline{\boldsymbol{X}}}_i = \frac{1}{n_i} \sum_{j \in s_i} \boldsymbol{x}_{ij}$, and $\overline{\boldsymbol{X}}_i = \frac{1}{N_i} \sum_{j \in U_i} \boldsymbol{x}_{ij}$.

## 5.3 MSE of the EBP

It is usually difficult to estimate the mean square prediction error(MSPE) of unit level model, which is caused by two reasons. First, the true distribution of error terms and non-sampled units in the unit level model is unknown, so it is impossible to obtain the density function for MSPE calculation. Second, even when the distribution of unsampled units is known, MSPE calculations are sometimes challenged by multiple integrals in the calculation of expectations due to the fact that model involve unit level data. In this paper, we use the parameter Bootstrap method mentioned in [18, 34] to estimate the MSPE.

The parametric bootstrap methods can be used to estimate the MSE of EBP $\widehat{\delta}_i^{EB}$ for finite populations. The method proceed as follows:

1. The robust estimation of parameters $\widehat{\boldsymbol{\beta}}, \widehat{\sigma}_v^2$ and $\widehat{\sigma}_e^2$ are obtained by using the DPD method mentioned in section 2;

2. Generate bootstrap domain effects as $v_i^* \overset{\text{iid}}{\sim} N(0, \widehat{\sigma}_v^2), i = 1, \ldots, m$; Generate, independently of $v_1^*, \ldots, v_m^*$, unit errors as $e_{ij}^* \overset{\text{ind}}{\sim} N(0, \widehat{\sigma}_e^2 k_{ij}^2), \quad j = 1, \ldots, N_i, \quad i = 1, \ldots, m$.

3. Generate a bootstrap population of response variables from the model

$$
y_{ij}^* = \mathbf{x}_{ij}' \widehat{\boldsymbol{\beta}} + v_i^* + e_{ij}^*, \quad j = 1, \ldots, N_i, \quad i = 1, \ldots, m.
$$

4. Let $\mathbf{y}_i^{P*} = (y_{i1}^*, \ldots, y_{iN_i}^*)^T$ denote the vector of generated bootstrap response variables for area $i$. Calculate target quantities for the bootstrap population as $\tau_i^* = h(\mathbf{y}_i^{P*}), i = 1, \ldots, m$.

5. Fit the model to the bootstrap data $\{(y_{ij}^*, \mathbf{x}_{ij}); j = 1, \ldots, N_i, i = 1, \ldots, m\}$ and obtain bootstrap model parameter estimators, denoted $\widehat{\sigma}_v^{2*}, \widehat{\sigma}_e^{2*}$, and $\widehat{\boldsymbol{\beta}}^*$.

6. Obtain the bootstrap EB estimator of $\tau_i$ using (18), denoted $\widehat{\tau}_i^{EB*}, i = 1, \ldots, m$.

7. Repeat steps (2)-(7) a large number of times $B$. Let $\tau_i^*(b)$ be true value and $\widehat{\tau}_i^{EB*}(b)$ the EB estimator obtained in $b$ th replicate of the bootstrap procedure, $b = 1, \ldots, B$.

8. The bootstrap MSE estimator of $\widehat{\tau}_i^{EB}$ is given by

$$
\text{mse}_B(\widehat{\tau}_i^{EB}) = B^{-1} \sum_{b=1}^{B} [\widehat{\tau}_i^{EB*}(b) - \tau_i^*(b)]^2.
$$

## 6 Application

In this section, we compare the effects of several robust Bayes estimators with the estimator proposed in this paper based on simulated and real data.

### 6.1 Simulation

**6.1.1 Contaminated distribution.**   In this paper, we do the same simulation as [18], that is, a unit level model with a single auxiliary variable $x$:

$$y_{ij} = \beta_0 + \beta_1 x_{ij} + v_i + e_{ij}, \quad i = 1, \ldots, k, \quad j = 1, \ldots, n$$

with $k$ = 40 and $n$ = 4. Where auxiliary variable $x_{ij} \sim N(1, 1)$, and the area-specific random effects $v_i$ and the random errors $e_{ij}$ were generated from the contaminated distribution $(1 - \eta)N(0, \sigma^2) + \eta N(0, \sigma_1^2)$. This means that a $(1 - \eta)$ proportion of the errors' were generated from the underlying "true" distribution $N(0, \sigma^2)$ and the remaining $\eta$ proportion of the errors were generated from the "arbitrary" contaminated distribution $N(0, \sigma_1^2)$. The choice $\eta = 0$ indicates no contamination of the distribution. For the underlying distributions, we set $\sigma_e^2 = \sigma_v^2 = 1$ and for the contaminated distributions, we set $\sigma_{e1}^2 = \sigma_{v1}^2 = 25$ and the proportion of contamination $\eta_1 = \eta_2 = 0.10$. We considered four possible combinations $\{(0, 0), (0, v), (0, e), (v, e)\}$ of contamination, where $(0, 0)$ indicates no contamination of the distributions, $(0, v)$ indicates the contamination only in the distribution of the area-specific random effects $v_i$, and so on.

For each simulation configuration, the regression coefficients were fixed at $(\beta_0, \beta_1)$ = (1, 1). We ran four sets of simulations, each of size 500. Given our focus on bias robustness, the main performance indicator for an MSE estimator in four studies is the relative bias, defined by

$$RABias = \frac{1}{m} \sum_{i=1}^{m} \frac{\left| \frac{1}{S} \sum_{j=1}^{S} \widehat{y}_{ij} - Y_i \right|}{Y_i}$$

Where the subscript $i$ indexes the small areas and the subscript $j$ indexes the $S$ Monte Carlo simulations, with $\widehat{y}_{ij}$ denoting the simulation $j$ value of the estimator in area $i$, and $Y_i$ denotes the actual vaule in area $i$. We also measured the stability of an MSE estimator by its relative MSE,

$$RAMSE = \frac{1}{m} \sum_{i=1}^{m} \frac{\left| \frac{1}{S} \sum_{j=1}^{S} (\widehat{y}_{ij} - Y_i)^2 \right|}{Y_i^2}.$$

In the simulation experiment, we compare the traditional maximum likelihood(ML) method, the robust estimation method(RML) mentioned in [18], and the robust minimum density power divergence method(RMD) proposed by us when taking different tuning parameters($\gamma$ = 0.1, 0.2, 0.3and the optimal $\gamma$ choosed by IWJ algorithm).

First we compare the estimates of model parameters under four contamination scenarios. Table 1 shows the RABiases and RAMSEs of estimators obtained from the robust estimation methods under different conditions, where the first row corresponding to each parameter represents the RABias and the second row represents the RAMSE.

The following conclusions can be clearly drawn from Table 1. In case of no contamination in the data, ML method in parameter estimation performance is best, However, RML and RMD methods with smaller tuning parameters are very similar to ML estimation results. This indicates that in this case, RMD method with smaller tuning parameters and RML method are almost as efficient as the ML method, and there is little difference between the RABias and

**Table 1. Simulated RABias and RAMSEs of robust and classical estimators of fixed effects and variance components.**

| Parameter | ML | RML | RMD1 | RMD2 | RMD3 | RMD.Opt. |
|---|---|---|---|---|---|---|
| No contamination | | $K=2$ | $\gamma=0.1$ | $\gamma=0.2$ | $\gamma=0.3$ | Optimal $\gamma$ |
| $\beta_0 = 1$ | 0.1423 | 0.1475 | 0.1458 | 0.1506 | 0.1567 | 0.1455 |
| | **0.0315** | 0.0339 | 0.0333 | 0.0362 | 0.0397 | 0.0328 |
| $\beta_1 = 1$ | 0.0729 | 0.0740 | 0.0735 | 0.0751 | 0.0778 | 0.0734 |
| | **0.0085** | 0.0089 | 0.0087 | 0.0091 | 0.0098 | 0.0087 |
| $\sigma_v^2 = 1$ | 0.2355 | 0.2455 | 0.2372 | 0.2460 | 0.2601 | 0.2367 |
| | **0.0882** | 0.0943 | 0.0899 | 0.0959 | 0.1058 | 0.0891 |
| $\sigma_e^2 = 1$ | 0.1096 | 0.1256 | 0.1110 | 0.1158 | 0.1230 | 0.1098 |
| | **0.0187** | 0.0241 | 0.0194 | 0.0212 | 0.0227 | 0.0191 |
| Contamination in $v$ | | | | | | |
| $\beta_0 = 1$ | 0.2449 | 0.2001 | 0.1785 | 0.1642 | 0.1661 | 0.1644 |
| | 0.0956 | 0.0650 | 0.0507 | 0.0428 | 0.0435 | **0.0426** |
| $\beta_1 = 1$ | 0.0053 | 0.0771 | 0.0745 | 0.0768 | 0.0796 | 0.0772 |
| | 0.0088 | 0.0095 | **0.0090** | 0.0095 | 0.0103 | 0.0096 |
| $\sigma_v^2 = 1$ | 2.4959 | 1.3465 | 0.9680 | 0.4691 | 0.3698 | 0.3578 |
| | 9.7369 | 2.7596 | 1.4408 | 0.3731 | 0.2360 | **0.2352** |
| $\sigma_e^2 = 1$ | 0.1085 | 0.1225 | 0.1125 | 0.1189 | 0.1260 | 0.1183 |
| | 0.0286 | 0.0233 | 0.0196 | 0.0218 | 0.0247 | **0.0204** |
| Contamination in $e$ | | | | | | |
| $\beta_0 = 1$ | 0.1789 | 0.1547 | 0.1606 | 0.1677 | 0.1760 | 0.1538 |
| | 0.0479 | 0.0374 | 0.0384 | 0.0423 | 0.0469 | **0.0271** |
| $\beta_1 = 1$ | 0.1296 | 0.0916 | 0.0993 | 0.0947 | 0.0971 | 0.0908 |
| | 0.0271 | 0.0132 | 0.0156 | 0.0141 | 0.0146 | **0.0127** |
| $\sigma_v^2 = 1$ | 0.3434 | 0.2687 | 0.2644 | 0.2687 | 0.2875 | 0.2642 |
| | 0.1828 | 0.1085 | 0.1081 | 0.1116 | 0.1285 | **0.1080** |
| $\sigma_e^2 = 1$ | 2.3794 | 0.5022 | 0.9472 | 0.4671 | 0.3394 | 0.3325 |
| | 6.4761 | 0.3009 | 1.0445 | 0.2873 | 0.1671 | **0.1668** |
| Contamination in $(v, e)$ | | | | | | |
| $\beta_0 = 1$ | 0.2464 | 0.2033 | 0.1830 | 0.1776 | 0.1841 | 0.1765 |
| | 0.0964 | 0.0674 | 0.0537 | 0.0503 | 0.0532 | **0.0502** |
| $\beta_1 = 1$ | 0.1314 | 0.0891 | 0.0921 | 0.0894 | 0.0918 | 0.0924 |
| | 0.0285 | 0.0125 | 0.0142 | 0.0126 | 0.0130 | **0.0122** |
| $\sigma_v^2 = 1$ | 2.407 | 1.3499 | 1.1223 | 0.5824 | 0.4833 | 0.4728 |
| | 9.5650 | 3.160 | 2.0507 | 0.6026 | 0.4054 | **0.3849** |
| $\sigma_e^2 = 1$ | 2.3847 | 0.5896 | 0.9997 | 0.4975 | 0.3735 | 0.3622 |
| | 6.5815 | 0.4072 | 1.1857 | 0.3272 | 0.1965 | **0.1868** |

[1]**ML**, maximum likelihood method; **RML**, robust maximum likelihood method of [18]; **RMD1, RMD2, RMD3**, robust minimum density power divergence method with tuning parameter $\gamma = 0.1, 0.2, 0.3$, respectively; **RMD.Opt**, robust minimum density power devergence method with tuning parameter obtained by optimal parameter selection algorithm.

RAMSE, while RMD method with larger tuning parameters performs poorly. It shows that the selection of tuning parameters is very important, and the optimal tuning parameters can be obtained according to the algorithm provided in Section 3.2. In the case of contamination in random effect $v_i$, The variance $\sigma$ estimated by the ML method has a large RABias and RAMSE, while the estimation by the RML method becomes smaller. However, the RMD method proposed by us is obviously better than the RML method, which has smaller bias and MSE in the

estimation of all parameters. According to the simulated data in the Table 1, when tuning parameter $\gamma$ was obtained by IWJ algorithm, the proposed RMD method provides better results for the estimation of all parameter. Similarly, in the case of outliers in the random errors $e_{ij}$, $\sigma_e^2$ estimated by ML method has a large bias and MSE. RML method has a good control on the influence of outliers, and the estimated bias and MSE of each parameter are relatively small. In the proposed RMD method, when $\gamma = 0.1$, the estimation of model variance in the results is reduced, but it is not as good as the RML method. However, when $\gamma > = 0.2$, the RMD method performs better than the RML method. In the case of both area effects $v_i$ and model errors $e_{ij}$ are contaminated, the ML estimator of variance component is heavily influenced by the outliers and produced much larger biases and MSE, RML method reduced the effects of outliers, but performance is not the best. The proposed RMD method is significantly better than the RML method, as we only need to select appropriate tuning parameters.

Next, using the same data set used in the above simulation, we consider the estimation of small area mean. The mean of the known auxiliary variables in the $i$th region are $\overline{X}_i = 1, i = 1, 2, \ldots, m$. Table 2 presents average simulated RABiases and average simulated RAMSE(averaged over the areas) of the estimators of small area means for the proposed and classical methods. The M-quantile(MQ) regression method is proposed by [17], and the area mean is

$$\hat{\bar{Y}}_i^{MQ} = N_i^{-1}\left\{\sum_{j\in S_i} y_j + (N_i - n_i)\hat{\bar{Y}}_{ri}\right\}$$

where, in this case, $\hat{\bar{Y}}_{ri} = \overline{\mathbf{x}}_{ri}\hat{\boldsymbol{\beta}}_{\hat{\tau}_i}$ with $\hat{\boldsymbol{\beta}}_{\hat{\tau}_i}$ estimating $\boldsymbol{\beta}_{\tau_i}$ in $q_{\tau_i}(y_j \mid \mathbf{x}_j) = \mathbf{x}_j\boldsymbol{\beta}_{\tau_i}$. the bias corrected version of the REBLUP(BC-RML) method is also used to compare with other methods, and the simulation results are shown in Table 2.

In the case of uncontaminated data, EBLUP obtained using the ML method appears to be the most efficient, as expected. The REBLUP using the RML method and the proposed robust MDPDE(RMDPDE) are also seen to be almost as efficient as the EBLUP. In the other three cases with outliers, we can get a conclusion consistent with the above simulation through the data in the Table 2, the small area mean obtained by ML method has a large bias and MSPE, the small area mean obtained by RML method performs slightly better, the proposed RMD method performs strictly better than the RML method.

In order to check the performance of several existing robust methods on the size of contaminated proportion and variance of contaminated distribution, we further simulated and verified the variation of MSE of estimated parameters with contaminated proportion and variance of contaminated distribution. Just like the above simulation steps, we consider the estimation

**Table 2. Simulated RABiases and RAMSE of robust and classical estimators of small area means (averaged over areas).**

| Contamination | ML | RML | MQ | BC-RML | RMD1 | RMD2 | RMD3 | RMD.Opt. |
|---|---|---|---|---|---|---|---|---|
| $(0, 0)$ | 0.0592 | 0.0375 | 0.0673 | 0.0865 | 0.1021 | 0.1038 | 0.1043 | 0.1020 |
| | **0.2114** | 0.2863 | 0.2172 | 0.2689 | 0.2347 | 0.2398 | 0.2491 | 0.2334 |
| $(v, 0)$ | 0.0219 | 0.1376 | 0.0352 | 0.0923 | 0.0917 | 0.0926 | 0.0903 | 0.0902 |
| | 1.0124 | 0.7765 | 0.7168 | 0.7342 | 0.6826 | 0.6636 | 0.6634 | **0.6583** |
| $(0, e)$ | 0.6247 | 0.4512 | 0.3298 | 0.3896 | 0.2921 | 0.2956 | 0.2978 | 0.2917 |
| | 3.1374 | 0.9376 | 0.7763 | 0.8864 | 0.1518 | 0.1496 | 0.1488 | **0.1312** |
| $(v, e)$ | 0.3384 | 0.2368 | 0.2245 | 0.1876 | 0.1436 | 0.1087 | 0.1078 | 0.1056 |
| | 4.0823 | 0.9856 | 0.8534 | 0.9438 | 0.2265 | 0.1183 | 0.1184 | **0.1087** |

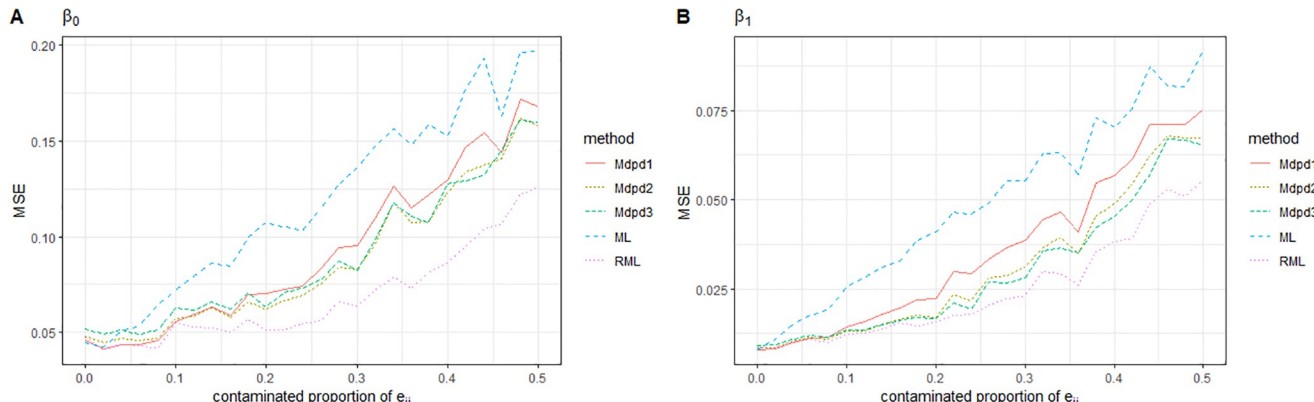

**Fig 1. The MSE of robust estimated parameters versus the contaminated proportion of $e_{ij}$.** Left-hand panel, $\beta_0$, right-hand panel, $\beta_1$. RML, the robust estimation method presented in [24]; ML, maximum likelihood estimation; Mdpd1, Mdpd2 and Mdpd3 represent the minimum density power devergence method with tuning parameter $\gamma = 0.1, 0.2, 0.3$ respectively.

effect under the three cases where the distribution of area-specific random error is contaminated, the distribution of unit random error is contaminated, and the distribution of both unit random error and area-specific random error is contaminated. In simulating contaminated data, we use the model in section 6.1.1 to generate data, and then perform parameter estimation using the method mentioned above. The simulation was repeated 500 times and the average MSE was taken into account to plot the change curve. In the first case, the MSE performance of the estimated parameters is considered as the contaminated proportion increases. The variance of contaminated distribution was fixed at 25, and the MSE of the estimated parameters under the three contamination scenarios was considered when the contaminated proportion changed between 0 and 0.5 with a step length of 0.02. In another case, MSE performance of the estimated parameters is considered when variance of contaminated distribution increases. In the contaminated distribution, the contaminated proportion was determined to be 0.1, considering that the variance of the contaminated distribution increased by 5 steps from 5 to 100, the change of MSE of the estimated parameters under the three contamination conditions was considered.

As can be seen from the Figs 1 and 2, when the contaminated proportion of random effect increases from 0 to 0.5, the MSE of the four parameters in the small area model increases with

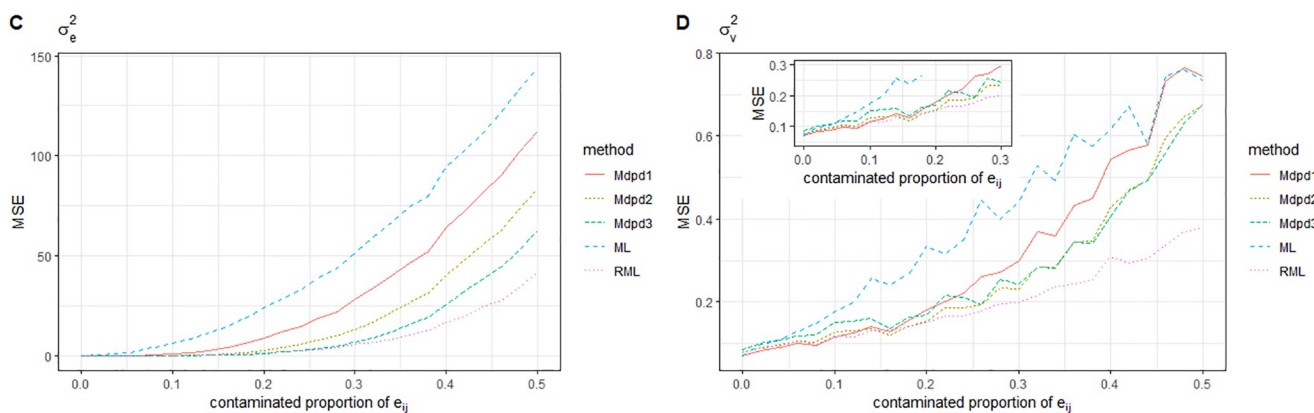

**Fig 2. The MSE of robust estimated parameters versus the contaminated proportion of $e_{ij}$.** Left-hand panel, $\sigma_e^2$, right-hand panel, $\sigma_v^2$.

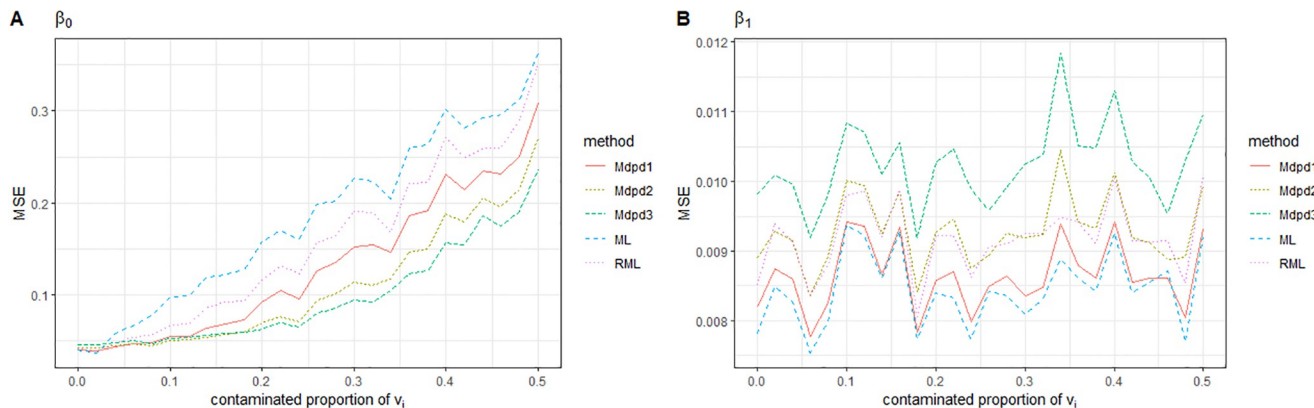

**Fig 3. The MSE of robust estimated parameters versus the contaminated proportion of $v_i$.** Left-hand panel, $\beta_0$, right-hand panel, $\beta_1$.

the increase of the contaminated proportion. Among the MSE of the four parameters, except for $\sigma_e^2$, the MSE of the other three parameters is not very large. As we expected, $\sigma_e^2$ is easily affected by contaminated proportion of $e_{ij}$, and the estimated MSE is relatively large. In the comparison between the several methods, ML performed worst. When the tuning parameter $\gamma$ was small(0–0.2), MDPD performed almost as well as RML. In some cases, MDPD performed better, but when the value of $\gamma$ was large, RML performed better.

Combined with Figs 3 and 4, we can easily find the following conclusions. When the area-specific random effect $v_i$ is contaminated, the MSE of $\beta_0, \sigma_v^2$ increases with the increase of the contaminated proportion, while the MSE of $\beta_1, \sigma_e^2$ is independent of the contaminated proportion, and the MSE is small. Comparing the performance of several methods for MSE estimation of four parameters, The MSE of $\beta_1, \sigma_e^2$ is small, and there is little difference among the methods. In the estimation of MSE of $\beta_0, \sigma_v^2$, ML method performs the worst, RML method performs slightly better than ML, and several MDPD methods perform significantly better than the above two methods. It further shows that the MDPD method is effective.

When both the random error $e_{ij}$ and the area-specific random error $v_i$ are contaminated, the change of MSE of the estimated parameters with the contaminated proportion is shown in Figs 5 and 6. As can be seen from the figure, when the contaminated proportion increases, the

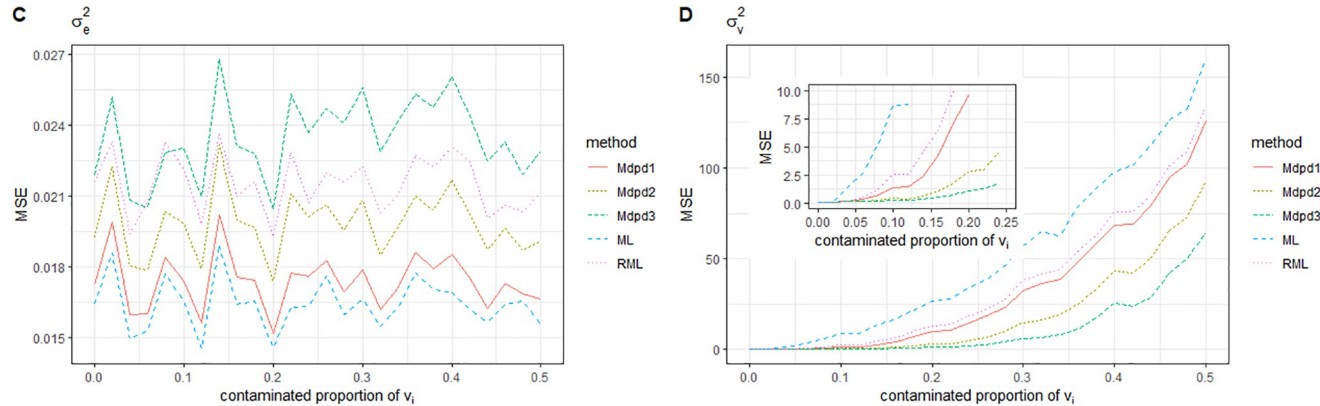

**Fig 4. Plot of the MSE of robust estimated parameters versus the contaminated proportion of $v_i$.** Left-hand panel, $\sigma_e^2$, right-hand panel, $\sigma_v^2$.

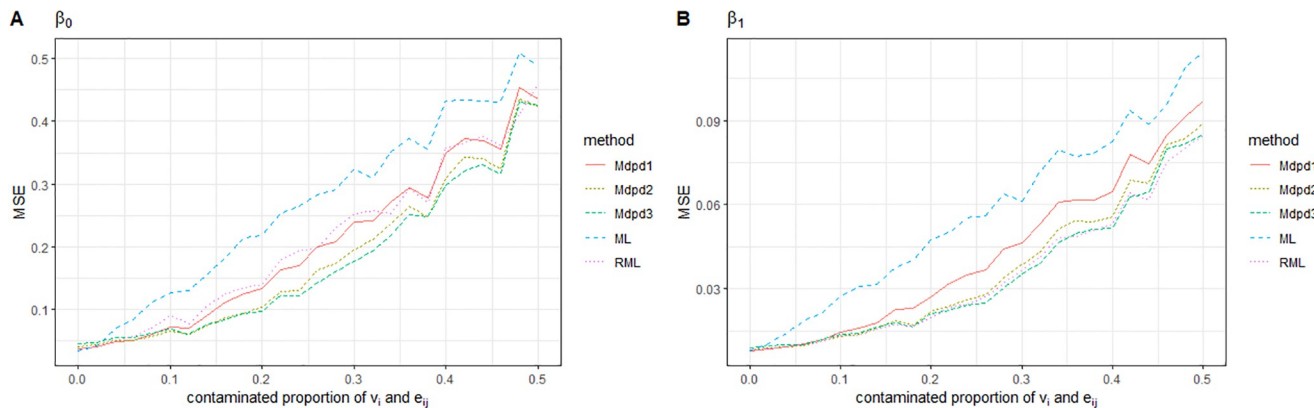

**Fig 5. The MSE of robust estimated parameters versus the contaminated proportion of ($v_i$, $e_{ij}$).** Left-hand panel, $\beta_0$, right-hand panel, $\beta_1$.

MSE of parameters also increases. In comparison with several estimation methods, the proposed MDPD method is obviously superior to RML and ML method, while ML method performs the worst.

When the individual error $e_{ij}$ is contaminated and the variance of the contamination distribution increases, the MSE of the four parameters is presented in Figs 7 and 8. As can be seen from the figure, when the variance of contamination distribution increases, only the MSE of parameters obtained by ML method shows a significant increase trend. However, the MSE of the parameters obtained by other robust estimation methods does not increase significantly, which indicates that the robust estimation method is uniformly effective in this case. Compared with several kinds of robust methods, the proposed MDPD method is superior to RML method, especially in the estimation of the MSE of $\sigma_e^2$.

When the area-specific random error is contaminated and the variance of the contamination distribution varies from 0 to 100, the MSE of the parameters is shown in the Figs 9 and 10. It can be seen from the figure that the MSE of $\beta_1$, $\sigma_e^2$ is independent of the variance of contamination distribution. In the estimation of MSE of parameters $\beta_0$, $\sigma_v^2$, ML method is seriously affected by the variance of contamination distribution. RML method has improved its performance, but not as good as MDPD method.

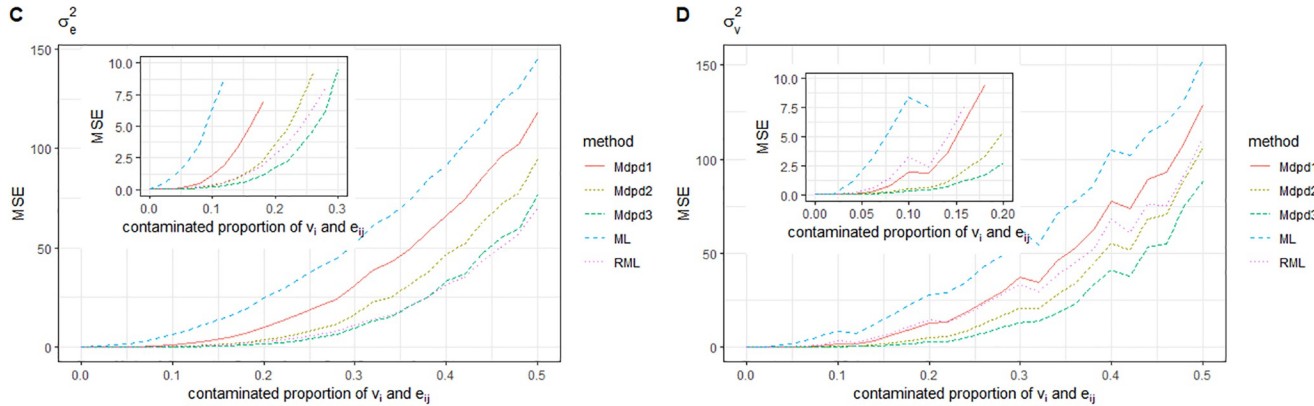

**Fig 6. The MSE of robust estimated parameters versus the contaminated proportion of ($v_i$, $e_{ij}$).** Left-hand panel, $\sigma_e^2$, right-hand panel, $\sigma_v^2$.

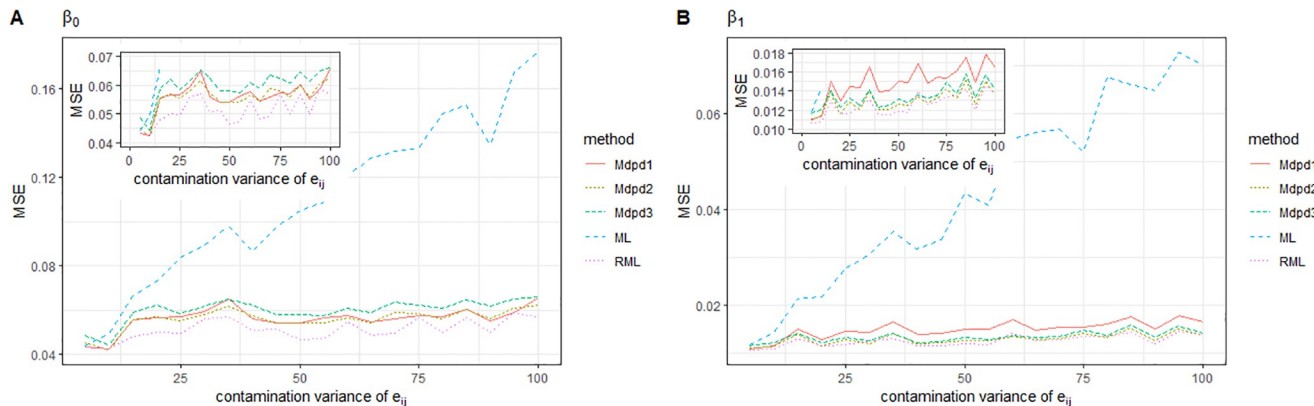

**Fig 7. The MSE of robust estimated parameters versus the contamination variance of $e_{ij}$.** Left-hand panel, $\beta_0$, right-hand panel, $\beta_1$.

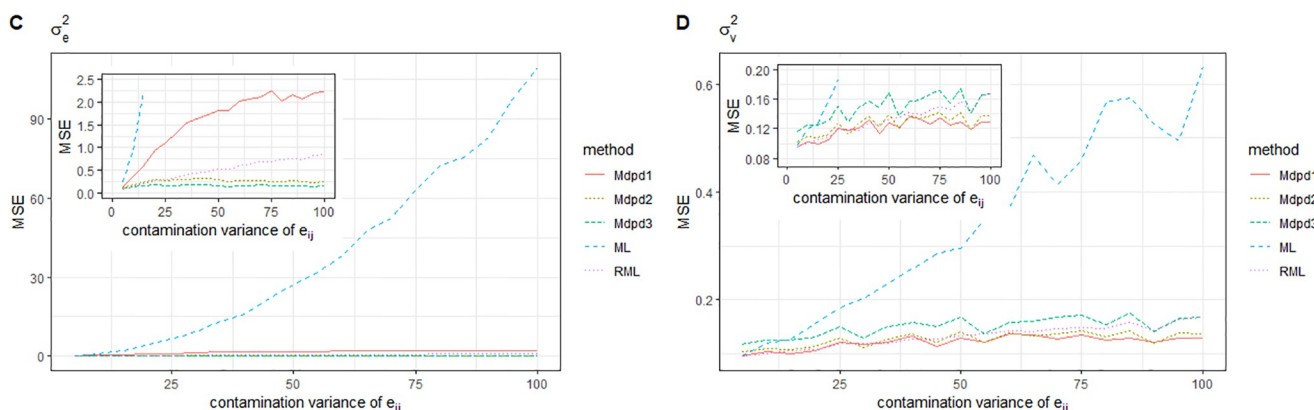

**Fig 8. The MSE of robust estimated parameters versus the contamination variance of $e_{ij}$.** Left-hand panel, $\sigma_e^2$, right-hand panel, $\sigma_v^2$.

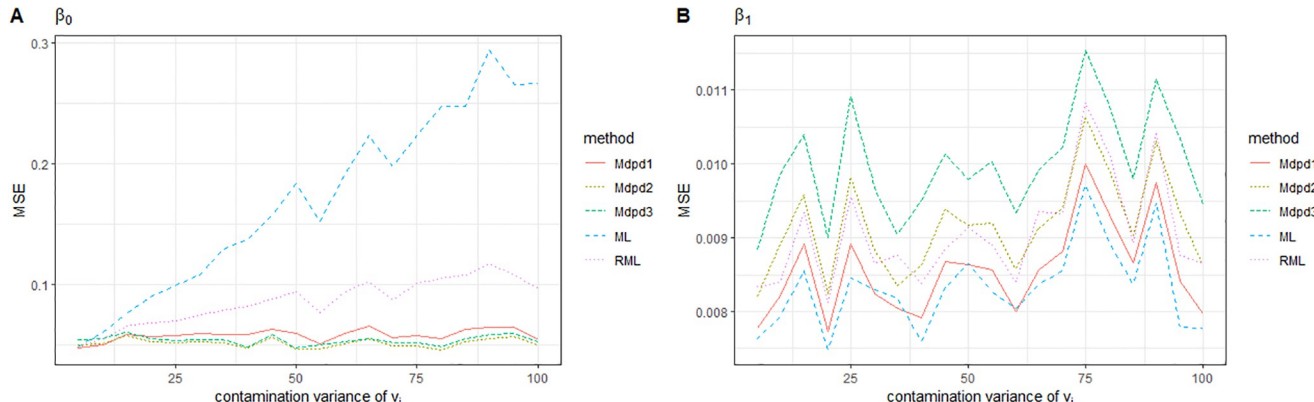

**Fig 9. The MSE of robust estimated parameters versus the contamination variance of $v_i$.** Left-hand panel, $\beta_0$, right-hand panel, $\beta_1$.

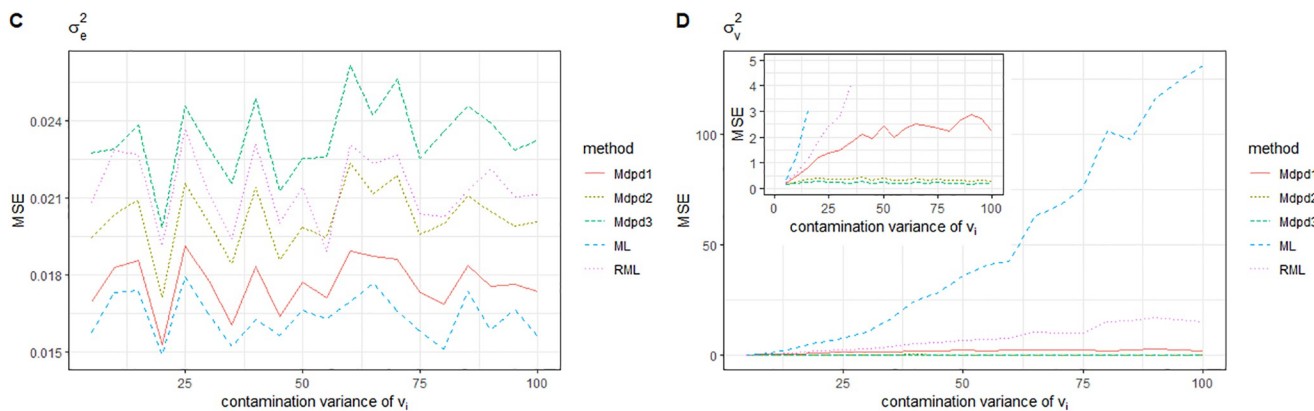

**Fig 10. The MSE of robust estimated parameters versus the contamination variance of $v_i$.** Left-hand panel, $\sigma_e^2$, right-hand panel, $\sigma_v^2$.

When both individual error $e_{ij}$ and area-specific random error $v_i$ are contaminated, Where the contaminated proportion is 0.1 and the variance of the contamination distribution varies from 0 to 100, the MSE of the parameters is shown in Figs 11 and 12. It can be seen from the figure that ML method has the worst performance, while RML method has improved the estimation effect, but it is not good for the estimation of parameters $\beta_0$, $\sigma_v^2$. In summary, MDPD method has good robustness for the estimation of several parameters.

**6.1.2 Finite population area means.** In this section, we focus on the small area means $\overline{Y}_i$ for finite population contains $m$ areas and the $i$th area of size $N_i$. We also use the method in 6.1.1 to simulate the performance of the population mean when $m = 40$ and each area is equal in size to $N_i = 40, 80, 200$ respectively. we generated a finite population using the unit-level model (2) with x $= (1, x)^t$, where $x \sim N(1, 1)$. For each of the four contamination schemes used in Section 6.1.1, we then generated a series of 500 population data sets. From each population data set, $ni = 4$ units from $N_i$ units in the $i$th area were selected as a random samples. For the data set of each simulation, we can use the robust method mentioned in the above simulation to obtain the small area mean $\overline{Y}_i$ for the $i$th area. Finally, we compare the average estimation of area mean after 500 simulations.

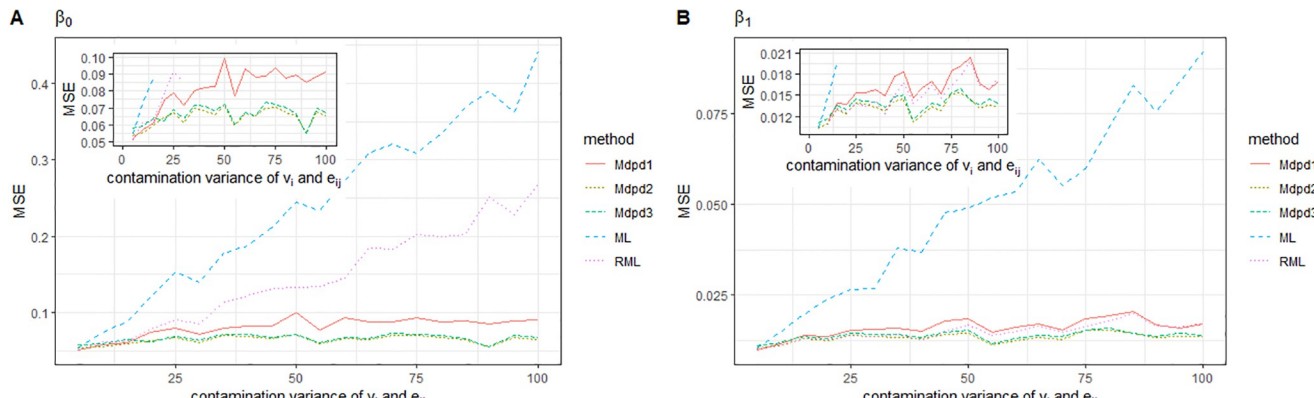

**Fig 11. The MSE of robust estimated parameters versus the contamination variance of $(v_i, v_{ij})$.** Left-hand panel, $\beta_0$, right-hand panel, $\beta_1$.

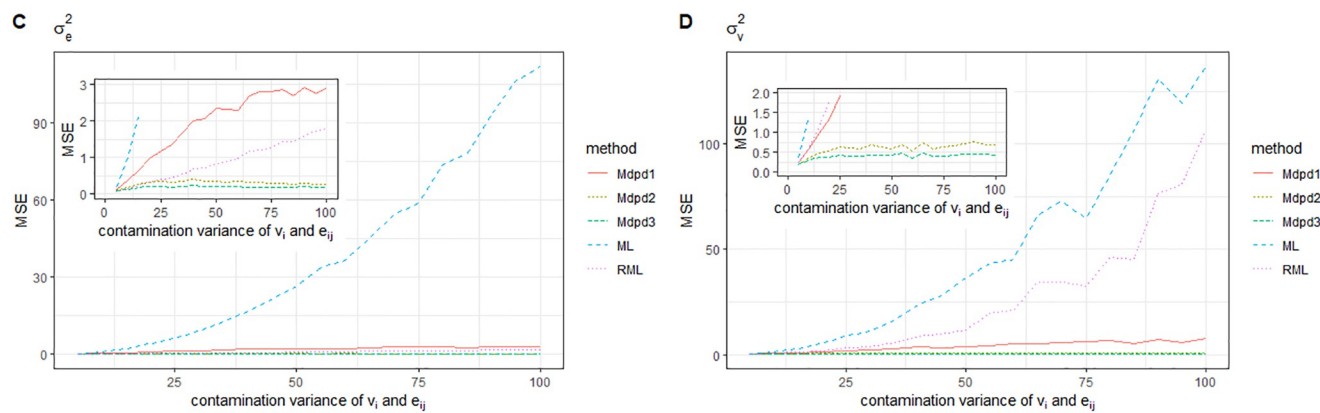

**Fig 12. The MSE of robust estimated parameters versus the contamination variance of $(v_i, e_{ij})$.** Left-hand panel, $\sigma_e^2$, right-hand panel, $\sigma_v^2$.

Table 3 presents average simulated RABiases(first line) and average simulated RAMSEs (second line,averaged over the areas) of the estimators of small area means $\overline{Y}_i$. From the simulation results, in general, the proposed RMD method has smaller MSE in most cases. In some cases, it is not as good as the estimated effect of RML method, but the difference is not

**Table 3. Simulated biases and mean squared errors of robust and classical estimators of fixed effects and variance components.**

| Contamination | ML | RML | MQ | BC-RML | RMD1 | RMD2 | RMD3 | RMD.Opt |
|---|---|---|---|---|---|---|---|---|
| $N_i=40$ | | $K=2$ | | | $\gamma=0.1$ | $\gamma=0.2$ | $\gamma=0.3$ | |
| $(0, 0)$ | 0.1694 | 0.2445 | 0.1696 | 0.2564 | 0.2378 | 0.2405 | 0.2433 | 0.2356 |
| | **0.1823** | 0.3134 | 0.1837 | 0.3129 | 0.2943 | 0.3018 | 0.3065 | 0.2905 |
| $(v, 0)$ | 0.1646 | 0.1254 | 0.1023 | 0.1178 | 0.2745 | 0.1178 | 0.1167 | 0.1155 |
| | 0.9793 | 0.6578 | 0.6481 | 0.6347 | 0.7015 | 0.6123 | 0.5901 | **0.4876** |
| $(0, e)$ | 0.3967 | 0.2359 | 0.2046 | 0.2314 | 0.1687 | 0.1598 | 0.1439 | 0.1376 |
| | 3.2674 | 1.1563 | 0.5842 | 0.5756 | 0.2643 | 0.2633 | 0.2823 | **0.1248** |
| $(v, e)$ | 0.3470 | 0.2919 | 0.1894 | 0.2167 | 0.0145 | 0.0065 | 0.0028 | 0.0023 |
| | 4.1672 | 1.2091 | 0.9348 | 0.9712 | 0.3721 | 0.2442 | 0.1386 | **0.1166** |
| $N_i=80$ | | | | | | | | |
| $(0, 0)$ | 0.1634 | 0.2419 | 0.1578 | 0.2322 | 0.1845 | 0.2217 | 0.2329 | 0.2313 |
| | **0.1765** | 0.3049 | 0.1807 | 0.2874 | 0.2187 | 0.2296 | 0.2467 | 0.2098 |
| $(v, 0)$ | 0.1625 | 0.1219 | 0.1015 | 0.1154 | 0.2045 | 0.1132 | 0.1095 | 0.1032 |
| | 0.9728 | 0.6429 | 0.6381 | 0.6352 | 0.6088 | 0.4571 | 0.3982 | **0.3011** |
| $(0, e)$ | 0.3912 | 0.2324 | 0.2011 | 0.2209 | 0.1674 | 0.1581 | 0.1329 | 0.1316 |
| | 3.2144 | 1.1035 | 0.5449 | 0.5568 | 0.2632 | 0.2561 | 0.2423 | **0.1183** |
| $(v, e)$ | 0.3304 | 0.2392 | 0.1874 | 0.2071 | 0.0135 | 0.0085 | 0.0038 | 0.0025 |
| | 3.8762 | 0.9218 | 0.8183 | 0.8232 | 0.3672 | 0.2418 | 0.1354 | **0.1146** |
| $N_i=200$ | | | | | | | | |
| $(0, 0)$ | 0.1521 | 0.2336 | 0.1494 | 0.1583 | 0.1642 | 0.1892 | 0.1980 | 0.1596 |
| | **0.1672** | 0.2598 | 0.1782 | 0.2332 | 0.2176 | 0.2345 | 0.2647 | 0.2065 |
| $(v, 0)$ | 0.1587 | 0.1125 | 0.0911 | 0.1092 | 0.1107 | 0.0934 | 0.0876 | 0.0861 |
| | 0.8831 | 0.4522 | 0.6037 | 0.6192 | 0.4081 | 0.3509 | 0.2214 | **0.1744** |
| $(0, e)$ | 0.4912 | 0.3393 | 0.2034 | 0.2267 | 0.1612 | 0.1431 | 0.1288 | 0.1102 |
| | 3.0913 | 0.9821 | 0.4376 | 0.4534 | 0.2421 | 0.2259 | 0.2110 | **0.1008** |
| $(v, e)$ | 0.3147 | 0.2317 | 0.1814 | 0.1986 | 0.0124 | 0.0078 | 0.0048 | 0.0032 |
| | 3.6591 | 0.8765 | 0.7021 | 0.7121 | 0.3616 | 0.2386 | 0.1211 | **0.1099** |

significant. In addition, we find that when the area size increases, such as $Ni = 200$, the estimation obtained by the proposed method has a smaller MSE, and the estimation effect is better than that obtained by traditional estimation method.

An interesting finding is that, when the area-specific random effects $v_i$ is contaminated, although the ML method has a large relative bias and MSE in the estimation of model parameters, the area mean obtained by the ML method has a smaller MSE. It shows that ML method is not sensitive to the variation of area-specific random effect, but is very sensitive to the variation of model random error $e_{ij}$. And the method proposed in this paper has a good robust effect on the two random effects.

## 6.2 Real data

In this section, we use the data that is used by [5] to estimate the area under corn and soybeans for each of $m = 12$ counties in North-Central Iowa. This data can be obtained from the R package "sae", which contains 37 samples of areas of corn and soybeans from the 12 counties, as well as the number of pixels classified by the LANDSAT satellite as corn and soybeans for each sample segment. The unit-level model was established with the data collected from farm interviews as the dependent variable and LANDSAT satellite data as the auxiliary variable.

$$y_{ij} = \beta_0 + \beta_1 x_{ij1} + \beta_2 x_{ij2} + v_i + \tilde{e}_{ij}, \tag{20}$$

which is a special case of model (1) with $k_{ij} = 1$, $\mathbf{x}_{ij} = (1, x_{ij1}, x_{ij2})^{\mathrm{T}}$, and $\boldsymbol{\beta} = (\beta_0, \beta_1, \beta_2)^{\mathrm{T}}$. Here, $y_{ij}$ is the number of hectares of corn (or soybeans), $x_{ij1}$ is the number of pixels classified as corn, and $x_{ij2}$ is the number of pixels classified as soybeans in the $j$th area segment of the $i$th county.

[5] identified an observation in Hardin county as an outlier, and they simply delete this observation when predicting the areas of corn and soybean. In [18], the robust estimation method is used to analyze this data, and the corresponding predicted value in the presence of outliers is given. Here, we use our proposed robust estimation method to model the data and analyze the influence of outlier on traditional estimators.

Considering the existence of outliers in the data, the robust estimation method proposed by us is considered to estimate and predicte the areas of corn in each segment. In addition, since there is only one outlier observation in this data, we select tuning parameter $\gamma = 0.01, 0.05$ for estimation in the proposed robust estimation method. In Table 4, regression coefficients and variance of random errors estimated by ML method, robust method proposed by [18] and MDPD method proposed by us are shown. The standard error for each parameter obtained from the asymptotic distribution in Section 4 is also shown in parentheses. When the tuning parameter $\gamma = 0.01$, It is clear from the table that the parameters estimated by MDPD are between those estimated by ML method and RML method. When the tuning parameter $\gamma$ was increased to 0.05, the coefficients estimated by the model changed significantly. By comparing

Table 4. Estimates of the model parameters from several methods. Standard errors are shown in the parenthesis.

| Coefficients | ML | RML | Mdpd1 | Mdpd2 |
|---|---|---|---|---|
| Intercept($\beta_0$) | 18.09(29.82) | 28.68(27.30) | 22.04 (14.44) | 37.64(14.18) |
| Corn pixels($\beta_1$) | 0.3657(0.0625) | 0.3545(0.0574) | 0.3614(0.0329) | 0.3441(0.0334) |
| Soybeans pixels($\beta_2$) | -0.0302(0.0650) | -0.0676(0.0596) | -0.0427(0.0337) | -0.0942(0.0323) |
| $\sigma_e^2$ | 280.2(71.55) | 213.7(63.94) | 274.3(66.84) | 243.11(68.38) |
| $\sigma_v^2$ | 47.80(56.51) | 113.7(64.57) | 52.64(50.32) | 75.01(69.94) |

**Table 5. Predicted mean hectares of corn per segment(bootstrap root MSPE in parentheses).**

| County | Sample segments | Estimated hectares | | | |
|---|---|---|---|---|---|
| | | EBLUP | RML | MDPD1 | MDPD2 |
| Cerro Gordo | 1 | 122.2(7.9) | 124.2(9.7) | 122.4(8.1) | 122.5(7.9) |
| Hamilton | 1 | 123.2(7.3) | 125.6(9.6) | 123.5(7.4) | 124.2(7.4) |
| Worth | 1 | 113.9(7.4) | 106.1 (9.3) | 113.6(7.1) | 112.0 (7.6) |
| Humboldt | 2 | 115.4(7.4) | 112.7 (8.7) | 115.0(6.8) | 112.6 (6.6) |
| Franklin | 3 | 136.1(7.0) | 143.2 (7.3) | 136.7(6.0) | 139.1 (6.0) |
| Pocahontas | 3 | 108.4(7.3) | 113.0 (7.0) | 108.5(6.0) | 109.0 (5.9) |
| Winnebago | 3 | 116.8(7.0) | 113.0 (7.5) | 116.8(6.0) | 115.8 (6.1) |
| Wright | 3 | 122.6(6.8) | 123.1 (7.5) | 122.5(6.0) | 121.8 (5.8) |
| Webster | 4 | 110.9(6.4) | 116.1 (6.6) | 111.1(5.4) | 112.0 (5.4) |
| Hancock | 5 | 124.4(5.9) | 122.6 (6.1) | 124.5(5.3) | 124.5 (5.4) |
| Kossuth | 5 | 113.4(5.9) | 105.2 (6.2) | 112.9(5.3) | 110.7 (5.4) |
| Hardin | 6 | 131.3(5.8) | 142.2 (6.0) | 131.5(5.4) | 132.3 (5.7) |

the standard errors of the parameters shown in the table, it can be seen that the parameters estimated by the proposed method have smaller standard errors.

In order to compare the estimation results, we show the predicted values of the mean hectares of corn per segment using the mldel (20). The EBLUP values obtained using the above estimation method are presented in the Table 5, where the Bootstrap estimates of the MSPE from 500 bootstrap samples are shown in parentheses. First of all, in terms of estimation results, the estimation of the region without outliers by using the proposed estimation method is closer to the result of ML estimation, and the prediction of region Hardin has been improved to some extent. Secondly, by comparing bootstrap MSPE, the MSPE values obtained by our proposed method are smaller, which shows that our proposed method is effective.

## 7 Discussion

In this paper, we propose a robust small area estimation method for unit level models with outlier observations. By introducing MDPD method, a robust estimation method with outliers and non-normal distribution errors is presented. Firstly, we have proposed an estimation equation for the parameters of the cell level model based on MDPD method and obtained the asymptotic properties of the model parameters. Secondly, combined with the asymptotic distribution of parameters, the selection procedure of optimal tuning parameter is given. Thirdly, the EBLUP values of unit and area mean in finite population is given. Finally, we verify the superior performance of our proposed method through simulated data and real data. In the simulation part, we simulate the robust estimation when the distribution is polluted, and discuss the effects of several kinds of robust estimation methods in three kinds of pollution cases. In particular, we discuss the variation of MSE of several estimation methods when the pollution ratio changes and the variance of pollution distribution changes. At the same time, the simulation results show that the proposed method can solve the outlier situation better. In the real data, we use the classical data of a small area estimation to illustrate the effectiveness of our proposed method. Through comparison, our proposed method can well deal with the special case of outlier observation.

Furthermore, it can be verified that the proposed method is also effective for random effects subject to other biased distributions. In this paper, we note that when the distribution of random errors is contaminated and the probability of contamination is greater than 0.3, the

performance of our proposed estimation method is generally poor than the robust estimation method in [18], but in this case, the MSE obtained by several kinds of methods are very large, and the robust estimation results are not very valuable. In the next step, the estimation method proposed by us can also be applied to the small region estimation problem of exponential distribution. Of course, these work need to be further studied and proved.

In this paper, we use IWJ algorithm to select the optimal tuning parameters. In similar related work, the parameter selection algorithm based on Hyvarinen score is given in [35]. In further research, this algorithm can be applied to the unit level model proposed in this paper, and the purpose of selecting the optimal tuning parameters can also be achieved. In addition, we only compared two classical robust estimation methods. In order to further illustrate the effectiveness of the method proposed in this paper, we can further compare the method proposed in this paper with [3, 17] and other methods.

## Supporting information

**S1 File. The proof of the theorem.**
(PDF)

## Acknowledgments

We are grateful Lanzhou University of Finance and Economics for providing with a learning and research platform.

## Author Contributions

**Conceptualization:** Zhiqiang Pang.

**Formal analysis:** Xijuan Niu.

**Methodology:** Zhaoxu Wang.

**Software:** Zhaoxu Wang.

**Supervision:** Zhiqiang Pang.

**Writing – original draft:** Zhaoxu Wang.

**Writing – review & editing:** Xijuan Niu.

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
