## [Decision Letter · Decision Letter 0]

2 Dec 2022

PONE-D-22-29623Robust Small Area Estimation for Unit Level Model with Density Power DivergencePLOS ONE

Dear Dr. Niu,

Thank you for submitting your manuscript to PLOS ONE. After careful consideration, we feel that it has merit but does not fully meet PLOS ONE’s publication criteria as it currently stands. Therefore, we invite you to submit a revised version of the manuscript that addresses the points raised during the review process.

As you can see, the reviewers, who are experts in the field of small are estimation, have major concerns about your article, and I agree with all their comments. Please make sure that the article is proof read before resubmitting it in order to improve the English. The simulation study needs to be expended substantially following reviewer 1 in particular. I also believe that the application needs to be improved given the scope of the journal; the data is too old. Please make sure the the software is released. 

We look forward to receiving your revised manuscript.

Kind regards,

Angelo Moretti, Ph.D.

Academic Editor

PLOS ONE

Journal Requirements:

"No"

5. Please upload a copy of Supporting Information Figure which you refer to in your text.

Reviewers' comments:

Reviewer's Responses to Questions

**Comments to the Author**

1. Is the manuscript technically sound, and do the data support the conclusions?

Reviewer #1: Partly

Reviewer #2: Yes

2. Has the statistical analysis been performed appropriately and rigorously? 

Reviewer #1: No

Reviewer #2: Yes

3. Have the authors made all data underlying the findings in their manuscript fully available?

Reviewer #1: Yes

Reviewer #2: Yes

4. Is the manuscript presented in an intelligible fashion and written in standard English?

Reviewer #1: No

Reviewer #2: Yes

5. Review Comments to the Author

Reviewer #1: The authors discussed a new methodology for dealing with outliers in the Small Area Estimation framework. The aim of the paper is attractive; however, there are many sentences with grammatical and spelling mistakes - which can all be corrected with a careful editing review of the manuscript. Moreover, there are also several aspects throughout the manuscript that are not so clear. In addition to the poor attention to the editing style (font size reduction from page 6), these aspects make the manuscript a little bit hard to read. Moreover, I think that a more concrete and recent application is necessary. In my opinion, the authors should also revise the simulation section.

Comments are in the attached file.

Reviewer #2: The authors proposed a robust small area estimation technique with unit-level models and density power divergence (DPD). The authors employs an objective function based on DPD applied to the marginal distribution of the observed values y_i and obtained asymptotic properties of the robust estimator. Furthermore, the authors derived a mean squared error estimator of the empirical best predictor. I think the paper is interesting and worth publishing. I have the following comments.

1. Estimation of random effects: While the model parameters are robustly estimated via DPD, the random effect prediction is derived in a standard way. As considered in Sugasawa (2020), is it possible to derive random effect estimator via Tweedie's formula?

2. Selection of the tuning parameter: although the authors adopted the selection method by Basak et al. (2021), using asymptotic mean squared errors. However, Sugasawa and Yonekura (2021) recently proposed a selection method based on the Hyvarinen score. This method is at least mentioned in the main text.

3. Simulation study: Under no contamination, the optimal value of gamma is ideally 0. However, the result indicates that the optimal value is considerably away from 0. Is there any reason or explanation for that?

4. Simulation study (line 323): The authors say ``According to the simulated data in the table 1, when tuning 323 parameter gamma = 0.026,... ", but I could not find the corresponding result in Table 1.

5. Table 3: It seems to show the results of bias and MSE of small area means according to the main text, but the caption indicates different results.

6. Real data: What is the optimal gamma here? Is there any reason for using only gamma=0.01 and 0.05 without selecting the optimal one?

7. I think the following two papers are closely related to the work, so it is worth mentioning in the main text (for example in Section 1).

- Kurisu et al. (2021) employed the gamma-divergence (similar to density power divergence) for the Fay-Herriot model and discussed empirical Bayes confidence intervals rather than MSE.

- Tang et al. (2018) proposed using global-local shrinkage priors for modeling random effects that allow potential outliers in the areal effect.

Reference

- Kurisu, D., T. Ishihara, and S. Sugasawa (2021). Adaptively robust small area estimation: Balancing robustness and efficiency of empirical Bayes confidence intervals. arXiv preprint arXiv:2108.11551.

- Sugasawa, S. and S. Yonekura (2021). On selection criteria for the tuning parameter in robust divergence. Entropy 23(9), 1147.

- Tang, X., M. Ghosh, N. S. Ha, and J. Sedransk (2018). Modeling random effects using global–local shrinkage priors in small area estimation. Journal of the American Statistical Association 113(524), 1476–1489.

6. PLOS authors have the option to publish the peer review history of their article (what does this mean?). If published, this will include your full peer review and any attached files.

Reviewer #1: No

Reviewer #2: No

---

## [Author Response · Author response to Decision Letter 0]

20 Apr 2023

Response to Reviewers

I am very glad to receive comments from reviewers, who have provided many meaningful suggestions for further improvement of this paper. In view of the amendments of the articles, this article made the following changes.

In view of the reviewer's main comments, this paper has been modified as follows:

1.For the first opinion, in the introduction part, the application background of DPD method is introduced in detail, and the advantages of this method compared with other methods are described, and some references are added. The changes in this section are between lines 90 and 108.

2.In section 5, the EBLUP can be obtained by substituting the estimated parameter values into the EBP, which is explained in section 5.

3.As for the third comments, the simulation part of the paper has been modified in detail. First, the Bias and MSE in the table become relative Bias and relative MSE. The parameter selection program is optimized and the values in the table are modified. This part is mainly in the numerical simulation part of the paper. It can be seen from the simulation results that the optimal adjustment parameters selected after modification can achieve the optimal estimation effect.

4.In the application of actual data, although the data used in this paper is relatively old, it can also illustrate the effect of the method proposed in this paper.

For minor modification suggestions, the main contents are as follows:

In line 14, the difference between direct and model-based estimates is explained.

Modify "area random effect and model random effect" to" individual and area-specific random effect".

In line 156, subheading 3.1 has been added

In the definition of DPD, the meaning of parameter θ is given.

Adjust the content of Section 5.3 to Section 3.2.

Formula 16 and 17 in line 280 are numbered

The section 5 explains the relationship between EBP and EBLUP.

In the simulation part, the expression formulas of relative bias deviation and relative MSE are given, and the performance results of relative MSE are shown in the simulation table.

Huberm methods have been changed to RML methods in the simulation part of the images and tables.

In the discussion section, the applicability of other parameter selection algorithms and other robust estimation methods is pointed out.

Some references have been added.

Some inaccuracies in grammar and expression have been corrected.

---

## [Decision Letter · Decision Letter 1]

16 May 2023

PONE-D-22-29623R1Robust Small Area Estimation for Unit Level Model with Density Power DivergencePLOS ONE

Dear Dr. Niu,

Thank you for submitting your manuscript to PLOS ONE. After careful consideration, we feel that it has merit but does not fully meet PLOS ONE’s publication criteria as it currently stands. Therefore, we invite you to submit a revised version of the manuscript that addresses the points raised during the review process.

I am happy to see that the article has been improved considerably. However, there is still some room for improvements. Could you please go over Reviewer 1's report carefully? I agree with the reviewer decision.  In particular, they asked for some comparisons which have not been carried out. I would like to see, at least, a careful discussion around that. Why did you decide not to provide such comparisons? Is there any literature available that you can include? Also, the issue related to the methods in Sinha and Rao needs to be addressed properly, given the importance of such paper in the literature. 

We look forward to receiving your revised manuscript.

Kind regards,

Angelo Moretti, Ph.D.

Academic Editor

PLOS ONE

Journal Requirements:

Reviewers' comments:

Reviewer's Responses to Questions

**Comments to the Author**

1. If the authors have adequately addressed your comments raised in a previous round of review and you feel that this manuscript is now acceptable for publication, you may indicate that here to bypass the “Comments to the Author” section, enter your conflict of interest statement in the “Confidential to Editor” section, and submit your "Accept" recommendation.

Reviewer #1: (No Response)

Reviewer #2: All comments have been addressed

2. Is the manuscript technically sound, and do the data support the conclusions?

Reviewer #1: Partly

Reviewer #2: Yes

3. Has the statistical analysis been performed appropriately and rigorously? 

Reviewer #1: Yes

Reviewer #2: Yes

4. Have the authors made all data underlying the findings in their manuscript fully available?

Reviewer #1: Yes

Reviewer #2: Yes

5. Is the manuscript presented in an intelligible fashion and written in standard English?

Reviewer #1: Yes

Reviewer #2: Yes

6. Review Comments to the Author

Reviewer #1: The Authors improved the manuscript. However, two of my questions remained unanswered:

- In particular, I have asked for a comparison in the simulation study with other methods, such as the MQ approach and the bias corrected version of the REBLUP, why the authors decided to not provide any comparison?

- the authors do not provide any code for running simulations; so my concern on why the results for variance of the random effects in two scenarios are different from the paper of Sinha and Rao remain unsolved,

- in the simulation section, are the measure in the Tables expressed in percentage? beacuse in the formula seems that they are not in percentage, but the values in the tables seems very high. Please clarify this point.

Reviewer #2: (No Response)

7. PLOS authors have the option to publish the peer review history of their article (what does this mean?). If published, this will include your full peer review and any attached files.

Reviewer #1: No

Reviewer #2: No

---

## [Author Response · Author response to Decision Letter 1]

15 Jun 2023

1.We have added a section on the MQ method and the REBLUP method for bias correction in the simulation experiments.

2.We have re-simulated the larger values in the tables using the same principle as Rao's method.

---

## [Decision Letter · Decision Letter 2]

3 Jul 2023

Robust Small Area Estimation for Unit Level Model with Density Power Divergence

PONE-D-22-29623R2

Dear Dr. Niu,

We’re pleased to inform you that your manuscript has been judged scientifically suitable for publication and will be formally accepted for publication once it meets all outstanding technical requirements.

Kind regards,

Angelo Moretti, Ph.D.

Academic Editor

PLOS ONE

Additional Editor Comments (optional):

Reviewers' comments:

Reviewer's Responses to Questions

**Comments to the Author**

1. If the authors have adequately addressed your comments raised in a previous round of review and you feel that this manuscript is now acceptable for publication, you may indicate that here to bypass the “Comments to the Author” section, enter your conflict of interest statement in the “Confidential to Editor” section, and submit your "Accept" recommendation.

Reviewer #1: All comments have been addressed

2. Is the manuscript technically sound, and do the data support the conclusions?

Reviewer #1: Yes

3. Has the statistical analysis been performed appropriately and rigorously? 

Reviewer #1: Yes

4. Have the authors made all data underlying the findings in their manuscript fully available?

Reviewer #1: Yes

5. Is the manuscript presented in an intelligible fashion and written in standard English?

Reviewer #1: Yes

6. Review Comments to the Author

Reviewer #1: I think that the authors have adequately addressed the comments in the revised version of the manuscript. Therefore, I have no further comments.

7. PLOS authors have the option to publish the peer review history of their article (what does this mean?). If published, this will include your full peer review and any attached files.

Reviewer #1: No

---

## [Editor Report · Acceptance letter]

7 Jul 2023

PONE-D-22-29623R2 

Robust Small Area Estimation for Unit Level Model with Density Power Divergence 

Dear Dr. Niu:

I'm pleased to inform you that your manuscript has been deemed suitable for publication in PLOS ONE. Congratulations! Your manuscript is now with our production department. 

Kind regards, 

on behalf of

Dr. Angelo Moretti 

Academic Editor

PLOS ONE